# Optimal Transport Perturbations for Safe Reinforcement Learning with Robustness Guarantees

**James Queeney**                                    *queeney@merl.com*
*Mitsubishi Electric Research Laboratories*

**Erhan Can Ozcan**                                    *cozcan@bu.edu*
*Division of Systems Engineering*
*Boston University*

**Ioannis Ch. Paschalidis**                          *yannisp@bu.edu*
**Christos G. Cassandras**                            *cgc@bu.edu*
*Department of Electrical and Computer Engineering and Division of Systems Engineering*
*Boston University*

**Reviewed on OpenReview:** *https://openreview.net/forum?id=cgSXpAR4Gl*

## Abstract

Robustness and safety are critical for the trustworthy deployment of deep reinforcement learning. Real-world decision making applications require algorithms that can guarantee robust performance and safety in the presence of general environment disturbances, while making limited assumptions on the data collection process during training. In order to accomplish this goal, we introduce a safe reinforcement learning framework that incorporates robustness through the use of an optimal transport cost uncertainty set. We provide an efficient implementation based on applying Optimal Transport Perturbations to construct worst-case virtual state transitions, which does not impact data collection during training and does not require detailed simulator access. In experiments on continuous control tasks with safety constraints, our approach demonstrates robust performance while significantly improving safety at deployment time compared to standard safe reinforcement learning.

## 1 Introduction

Deep reinforcement learning (RL) is a data-driven framework for sequential decision making that has demonstrated the ability to solve complex tasks, and represents a promising approach for improving real-world decision making (Dulac-Arnold et al., 2021). In order for deep RL to be trusted for deployment in real-world decision making settings, however, robustness and safety are of the utmost importance (Xu et al., 2022). As a result, techniques have been developed to incorporate both robustness and safety into deep RL. Robust RL methods protect against worst-case transition models in an uncertainty set (Nilim & Ghaoui, 2005; Iyengar, 2005), while safe RL methods incorporate safety constraints into the training process (Altman, 1999).

In real-world applications, disturbances in the environment can take many forms that are difficult to model in advance. Therefore, we require methods that can guarantee robust performance and safety under general forms of environment uncertainty. Unfortunately, popular approaches to robustness in deep RL consider very structured forms of uncertainty in order to facilitate efficient implementations. Adversarial methods implement a specific type of perturbation, such as the application of a physical force (Pinto et al., 2017) or a change in the action that is deployed (Tessler et al., 2019a). Parametric approaches, on the other hand, consider robustness with respect to environment characteristics that can be altered in a simulator (Rajeswaran et al., 2017; Peng et al., 2018; Mankowitz et al., 2020). When we lack domain knowledge on the structure of potential disturbances, these techniques may not guarantee robust performance and safety.

**Worst-case transition models**  **Optimal Transport Perturbations**

Figure 1: Illustration of Optimal Transport Perturbations. Left: Worst-case transition models in the optimal transport uncertainty set $\mathcal{P}_{s,a} \subseteq P(\mathcal{S})$ that correspond to the robust Bellman operators in (5) and (6). Right: Tractable reformulation by applying Optimal Transport Perturbations directly in $\mathcal{S}$ to a given next state sample $\hat{s}' \sim \hat{p}_{s,a}$. The black arrow denotes the state transition observed in the nominal environment, and the dashed arrows denote virtual state transitions used only to calculate robust Bellman operators.

Another drawback of existing approaches is their need to directly modify the environment during training. Parametric methods assume the ability to generate a range of training environments with a detailed simulator, while adversarial methods directly influence the data collection process by attempting to negatively impact performance. In applications where simulators are inaccurate or unavailable, however, parametric methods cannot be applied and real-world data collection may be required for training. In this context, it is also undesirable to implement adversarial perturbations while interacting in the environment. Therefore, in many real-world domains, we must consider alternative methods for learning safe policies with robustness guarantees.

In this work, we introduce a safe RL framework that provides robustness to general forms of environment disturbances using standard data collection in a *nominal training environment*. We consider robustness over a general uncertainty set defined using the optimal transport cost between transition models, and we show how our framework can be efficiently implemented by applying *Optimal Transport Perturbations* after data collection to construct worst-case *virtual* state transitions. See Figure 1 for an illustration. These perturbations can be added to the training process of many popular safe RL algorithms to incorporate robustness to unknown disturbances, without harming performance during training or requiring access to a range of simulated training environments.

We summarize our contributions as follows:

1. We formulate a safe RL framework that incorporates robustness to general disturbances using the optimal transport cost between transition models.

2. We show in Theorem 1 that the resulting worst-case optimization problems over transition models can be reformulated as adversarial perturbations to state transitions in the training environment.

3. We propose an efficient deep RL implementation of our Optimal Transport Perturbations, which we use to construct worst-case virtual state transitions *without impacting data collection during training*.

4. We demonstrate that the use of Optimal Transport Perturbations leads to robust performance and safety both during training and in the presence of disturbances through experiments on continuous control tasks with safety constraints in the Real-World RL Suite (Dulac-Arnold et al., 2020; 2021).

## 2 Related work

### 2.1 Safe reinforcement learning

The most common approach to modeling safety in RL is to incorporate constraints on expected total costs in a Constrained Markov Decision Process (Altman, 1999), which is the definition of safety we consider in this work. In recent years, several deep RL algorithms have been developed for this framework. A popular approach is to solve the Lagrangian relaxation of the constrained problem (Tessler et al., 2019b; Ray et al., 2019; Stooke et al., 2020), which is supported by theoretical results establishing that constrained RL has zero duality gap (Paternain et al., 2019). Other approaches to safe RL construct closed-form solutions to guide policy updates (Achiam et al., 2017; Liu et al., 2022), or consider immediate switching between the objective and constraints to better satisfy safety during training (Xu et al., 2021).

A separate line of work focuses on the issue of safe exploration during data collection (Brunke et al., 2022). Control-theoretic approaches focus on avoiding sets of unsafe states through the use of control barrier functions (Cheng et al., 2019; Emam et al., 2021; Ma et al., 2021) or other safety filters (Dalal et al., 2018), but often require additional assumptions on system dynamics. Recently, action correction mechanisms have been applied with unknown dynamics through the use of learned safety critics (Srinivasan et al., 2020; Bharadhwaj et al., 2021) and recovery policies (Thananjeyan et al., 2021; Wagener et al., 2021). We do not directly consider the issue of safe exploration in this work, but it is possible to combine our robust training approach with correction mechanisms for safe exploration.

### 2.2 Robust reinforcement learning

Robust RL methods account for uncertainty in the environment by considering worst-case transition models from an uncertainty set (Nilim & Ghaoui, 2005; Iyengar, 2005). In order to scale the robust RL framework to the deep RL setting, most techniques have focused on (i) parametric uncertainty or (ii) adversarial training.

Parametric uncertainty methods generate multiple training environments by modifying parameters in a simulator that are often determined based on domain knowledge. Domain randomization (Tobin et al., 2017; Peng et al., 2018) maximizes average performance over a range of training environments, which has been referred to as a soft-robust approach (Derman et al., 2018). Other methods directly impose a robust perspective towards parametric uncertainty by focusing on the worst-case training environments generated over a range of simulator parameters (Rajeswaran et al., 2017; Abdullah et al., 2019; Mankowitz et al., 2020). All of these approaches assume access to a simulated version of the real environment, as well as the ability to modify parameters of this simulator.

Adversarial RL methods represent an alternative approach to robustness that introduce perturbations directly into the training process. In order to learn policies that perform well under worst-case disturbances, these perturbations are trained to minimize performance. Deep RL approaches to adversarial training have introduced perturbations in the form of physical forces in the environment (Pinto et al., 2017), as well as adversarial corruptions to actions (Tessler et al., 2019a; Vinitsky et al., 2020) and state observations (Mandlekar et al., 2017; Zhang et al., 2020; Kuang et al., 2022; Liu et al., 2023a;b). In this work, we learn adversarial perturbations on state transitions, but different from most adversarial RL methods we apply these perturbations to construct virtual transitions *without impacting the data collection process.*

Finally, safety and robustness have recently been considered together in a unified RL framework. Mankowitz et al. (2021) and Russel et al. (2021) propose a formulation that incorporates robustness into both the objective and constraints in safe RL. We consider this general framework as a starting point for our work.

## 3 Preliminaries

### 3.1 Safe reinforcement learning

Consider an infinite-horizon, discounted Constrained Markov Decision Process (C-MDP) (Altman, 1999) defined by the tuple $(\mathcal{S}, \mathcal{A}, p, r, c, \rho_0, \gamma)$, where $\mathcal{S}$ is the set of states, $\mathcal{A}$ is the set of actions, $p : \mathcal{S} \times \mathcal{A} \to P(\mathcal{S})$

is the transition model where $P(\mathcal{S})$ denotes the space of probability measures over $\mathcal{S}$, $r : \mathcal{S} \times \mathcal{A} \to \mathbb{R}$ is the reward function, $c : \mathcal{S} \times \mathcal{A} \to \mathbb{R}$ is the cost function, $\rho_0$ is the initial state distribution, and $\gamma$ is the discount rate.

We model the agent's decisions as a stationary policy $\pi : \mathcal{S} \to P(\mathcal{A})$. For a given C-MDP and policy $\pi$, the expected total discounted rewards and costs are given by $J_{p,r}(\pi) = \mathbb{E}_{\tau \sim (\pi, p)} \left[ \sum_{t=0}^{\infty} \gamma^t r(s_t, a_t) \right]$ and $J_{p,c}(\pi) = \mathbb{E}_{\tau \sim (\pi, p)} \left[ \sum_{t=0}^{\infty} \gamma^t c(s_t, a_t) \right]$, respectively, where $\tau \sim (\pi, p)$ represents a trajectory sampled according to $s_0 \sim \rho_0$, $a_t \sim \pi(\cdot \mid s_t)$, and $s_{t+1} \sim p(\cdot \mid s_t, a_t)$. The goal of safe RL is to find a policy $\pi$ that solves the constrained optimization problem

$$\max_{\pi} \ J_{p,r}(\pi) \quad \text{s.t.} \quad J_{p,c}(\pi) \leq B, \tag{1}$$

where $B$ is a safety budget on expected total discounted costs.

We denote the state-action value functions (i.e., Q functions) of $\pi$ for a given C-MDP as $Q_{p,r}^{\pi}(s,a)$ and $Q_{p,c}^{\pi}(s,a)$. Off-policy optimization techniques (Xu et al., 2021; Liu et al., 2022) iteratively optimize (1) by considering the related optimization problem

$$\max_{\pi} \ \mathbb{E}_{s \sim \mathcal{D}} \left[ \mathbb{E}_{a \sim \pi(\cdot|s)} \left[ Q_{p,r}^{\pi_k}(s,a) \right] \right] \quad \text{s.t.} \quad \mathbb{E}_{s \sim \mathcal{D}} \left[ \mathbb{E}_{a \sim \pi(\cdot|s)} \left[ Q_{p,c}^{\pi_k}(s,a) \right] \right] \leq B, \tag{2}$$

where $\pi_k$ is the current policy and $\mathcal{D}$ is a replay buffer containing data collected during training.

### 3.2 Robust and safe reinforcement learning

We are often interested in finding a policy $\pi$ that achieves strong, safe performance across a range of related environments. In order to accomplish this, Mankowitz et al. (2021) and Russel et al. (2021) propose a Robust Constrained MDP (RC-MDP) framework defined by the tuple $(\mathcal{S}, \mathcal{A}, \mathcal{P}, r, c, \rho_0, \gamma)$, where $\mathcal{P}$ represents an uncertainty set of transition models. We assume $\mathcal{P} = \bigotimes_{(s,a) \in \mathcal{S} \times \mathcal{A}} \mathcal{P}_{s,a}$, where $\mathcal{P}_{s,a}$ is a set of transition models $p_{s,a} = p(\cdot \mid s, a) \in P(\mathcal{S})$ at a given state-action pair and $\mathcal{P}$ is the product of these sets. This structure is referred to as rectangularity, and is a common assumption in the literature (Nilim & Ghaoui, 2005; Iyengar, 2005).

In order to learn a policy with robust performance and safety across all environments in $\mathcal{P}$, the RC-MDP framework considers a robust version of (1) given by

$$\max_{\pi} \ \inf_{p \in \mathcal{P}} J_{p,r}(\pi) \quad \text{s.t.} \quad \sup_{p \in \mathcal{P}} J_{p,c}(\pi) \leq B. \tag{3}$$

**Remark 1.** *A policy $\pi$ that solves* (3) *satisfies the safety constraint across all transition models in $\mathcal{P}$ (i.e., achieves robust safety). Within this set of feasible policies, a policy $\pi$ that solves* (3) *maximizes the worst-case performance across all transition models in $\mathcal{P}$ (i.e., achieves robust performance).*

We are interested in solving (3) to find a policy $\pi$ with robust performance and safety as described in Remark 1. This is in contrast to solving the standard safe RL problem in (1), which only considers performance and safety in a single environment. As in the standard safe RL setting, we can apply off-policy optimization techniques to iteratively optimize (3) by considering the related optimization problem

$$\max_{\pi} \ \mathbb{E}_{s \sim \mathcal{D}} \left[ \mathbb{E}_{a \sim \pi(\cdot|s)} \left[ Q_{\mathcal{P},r}^{\pi_k}(s,a) \right] \right] \quad \text{s.t.} \quad \mathbb{E}_{s \sim \mathcal{D}} \left[ \mathbb{E}_{a \sim \pi(\cdot|s)} \left[ Q_{\mathcal{P},c}^{\pi_k}(s,a) \right] \right] \leq B, \tag{4}$$

where $Q_{\mathcal{P},r}^{\pi}(s,a)$ and $Q_{\mathcal{P},c}^{\pi}(s,a)$ represent robust Q functions. In this work, we focus on how to efficiently learn the robust Q functions that are needed for the robust and safe policy update in (4).

### 3.3 Robust Bellman operators

Compared to the standard safe RL update in (2), the only difference in the robust and safe RL update of (4) comes from the use of robust Q functions. Therefore, in order to incorporate robustness into existing deep

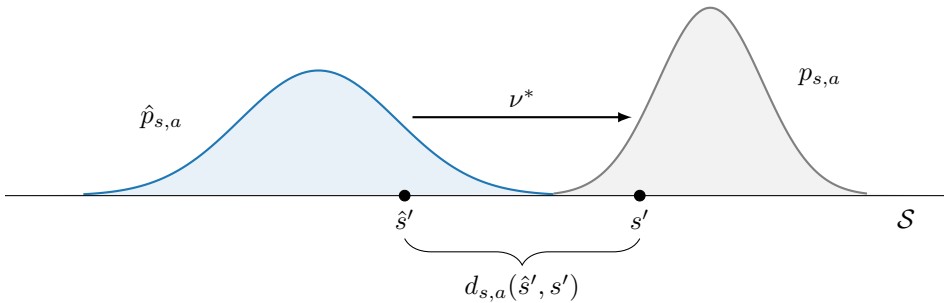

Figure 2: Illustration of optimal transport cost between transition models $\hat{p}_{s,a}, p_{s,a} \in P(\mathcal{S})$. Informally, $\text{OTC}_{d_{s,a}}(\hat{p}_{s,a}, p_{s,a})$ represents the cost of transporting the probability mass of $\hat{p}_{s,a}$ to $p_{s,a}$ using the optimal (i.e., minimum cost) transport plan $\nu^*$, where the transport cost is determined by $d_{s,a}$.

safe RL algorithms, we must be able to efficiently learn $Q_{\mathcal{P},r}^\pi(s,a)$ and $Q_{\mathcal{P},c}^\pi(s,a)$. These robust Q functions represent the unique fixed points of the corresponding robust Bellman operators (Nilim & Ghaoui, 2005; Iyengar, 2005)

$$\mathcal{T}_{\mathcal{P},r}^\pi Q_r(s,a) := r(s,a) + \gamma \inf_{p_{s,a} \in \mathcal{P}_{s,a}} \mathbb{E}_{s' \sim p_{s,a}} \left[ V_r^\pi(s') \right], \tag{5}$$

$$\mathcal{T}_{\mathcal{P},c}^\pi Q_c(s,a) := c(s,a) + \gamma \sup_{p_{s,a} \in \mathcal{P}_{s,a}} \mathbb{E}_{s' \sim p_{s,a}} \left[ V_c^\pi(s') \right], \tag{6}$$

where we write $V_r^\pi(s') = \mathbb{E}_{a' \sim \pi(\cdot|s')} \left[ Q_r(s',a') \right]$ and $V_c^\pi(s') = \mathbb{E}_{a' \sim \pi(\cdot|s')} \left[ Q_c(s',a') \right]$. Note that $\mathcal{T}_{\mathcal{P},r}^\pi$ and $\mathcal{T}_{\mathcal{P},c}^\pi$ are contraction operators (Nilim & Ghaoui, 2005; Iyengar, 2005), so we can apply standard temporal difference learning techniques to learn $Q_{\mathcal{P},r}^\pi(s,a)$ and $Q_{\mathcal{P},c}^\pi(s,a)$. In order to do so, we must be able to calculate the Bellman targets in (5) and (6), which involve worst-case optimization problems over transition models that depend on the choice of uncertainty set $\mathcal{P}_{s,a}$ at every state-action pair. In order to estimate these Bellman targets, popular choices of $\mathcal{P}_{s,a}$ in the literature require the ability to change physical parameters of the environment (Peng et al., 2018) or directly apply adversarial perturbations during trajectory rollouts (Tessler et al., 2019a) to calculate worst-case transitions. However, because these implementations rely on multiple simulated training environments or potentially dangerous adversarial interventions, they are not compatible with settings that require real-world data collection for training.

## 4 Optimal transport uncertainty set

In this work, we consider an uncertainty set based on optimal transport cost, and we show that this choice of uncertainty set leads to an efficient implementation of the worst-case optimization problems in (5) and (6). In order to do so, we assume that $\mathcal{S}$ is a Polish space (i.e., a separable, completely metrizable topological space). Note that the Euclidean space $\mathbb{R}^n$ is Polish, so this is not very restrictive. Next, we define $\mathcal{P}_{s,a}$ using the optimal transport cost between transition models.

**Definition 1** (Optimal transport cost). *Let $\mathcal{S}$ be a Polish space, and let $d_{s,a} : \mathcal{S} \times \mathcal{S} \to \mathbb{R}_+$ be a non-negative, lower semicontinuous function satisfying $d_{s,a}(s',s') = 0$ for all $s' \in \mathcal{S}$. Then, the optimal transport cost between two transition models $\hat{p}_{s,a}, p_{s,a} \in P(\mathcal{S})$ is defined as*

$$\text{OTC}_{d_{s,a}}(\hat{p}_{s,a}, p_{s,a}) = \inf_{\nu \in \Gamma(\hat{p}_{s,a}, p_{s,a})} \int_{\mathcal{S} \times \mathcal{S}} d_{s,a}(\hat{s}', s') \mathrm{d}\nu(\hat{s}', s'),$$

*where $\Gamma(\hat{p}_{s,a}, p_{s,a})$ is the set of all couplings of $\hat{p}_{s,a}$ and $p_{s,a}$.*

See Figure 2 for an illustration of Definition 1. For a given transport cost function $d_{s,a}$, $\text{OTC}_{d_{s,a}}(\hat{p}_{s,a}, p_{s,a})$ represents the minimum cost required to transport the probability mass of $\hat{p}_{s,a}$ to $p_{s,a}$, and the optimal coupling $\nu^*$ describes the transport plan that achieves this minimum cost. If $d_{s,a}$ is chosen to be a metric

raised to some power $p \geq 1$, we recover the $p$-Wasserstein distance raised to the power $p$ as a special case (Chen & Paschalidis, 2020). If we let $d_{s,a}(\hat{s}', s') = \mathbf{1}_{\hat{s}' \neq s'}$, we recover the total variation distance as a special case (Villani, 2008).

By considering the optimal transport cost from some nominal transition model $\hat{p}_{s,a}$, we define the optimal transport uncertainty set as follows.

**Definition 2** (Optimal transport uncertainty set). *For a given nominal transition model $\hat{p}_{s,a}$, transport cost function $d_{s,a}$, and radius $\epsilon_{s,a}$ at $(s,a) \in \mathcal{S} \times \mathcal{A}$, the* optimal transport uncertainty set *is defined as*

$$\mathcal{P}_{s,a} = \left\{ p_{s,a} \in P(\mathcal{S}) \mid \mathrm{OTC}_{d_{s,a}}(\hat{p}_{s,a}, p_{s,a}) \leq \epsilon_{s,a} \right\}.$$

This uncertainty set has previously been considered in robust RL for the special case of the Wasserstein distance (Abdullah et al., 2019; Hou et al., 2020; Kuang et al., 2022). As we will show in the following sections, the use of an optimal transport uncertainty set leads to an efficient, model-free implementation of robust and safe RL *that only requires the ability to collect data in a nominal environment.*

Note that the optimal transport uncertainty set in Definition 2 can be applied across a variety of domains. Optimal transport cost remains valid for distributions that do not share the same support, unlike other popular measures between distributions such as the Kullback-Leibler divergence. As a result, the optimal transport uncertainty set is very general and can be applied to both stochastic and deterministic transition models. Optimal transport cost also accounts for the relationship between states in $\mathcal{S}$ through the function $d_{s,a}$, and allows significant flexibility in the choice of $d_{s,a}$. This includes threshold-based binary comparisons (Pydi & Jog, 2020) and percentage-based comparisons between states that are not metrics or pseudo-metrics. To demonstrate the flexibility of our approach for the common setting where $\mathcal{S} = \mathbb{R}^n$, we consider in our experiments a percentage-based comparison of state transitions given by

$$d_{s,a}(\hat{s}', s') = \frac{1}{n} \sum_{i=1}^{n} \left( \frac{s'_i - s_i}{\hat{s}'_i - s_i} - 1 \right)^2, \tag{7}$$

with the convention that $0/0 = 1$ so that $d_{s,a}(s', s') = 0$ for all $s' \in \mathcal{S}$. Note that (7) satisfies all requirements in Definition 1, and is not a metric or pseudo-metric. In more general settings, we can choose $d_{s,a}$ based on the application to reflect the geometry of $\mathcal{S}$ in a meaningful way.

## 5 Reformulation as worst-case virtual state transitions

We are interested in solving the robust and safe RL problem in (3) for an optimal transport uncertainty set, which we can accomplish by applying the policy updates given by (4). In order to learn the robust Q functions that appear in (4) for an optimal transport uncertainty set, we focus on how to efficiently calculate the robust Bellman operators in (5) and (6). We consider the following main assumptions.

**Assumption 1.** *For any $\pi$ and $Q_r(s', a')$ in (5), $V_r^\pi(s') = \mathbb{E}_{a' \sim \pi(\cdot|s')}[Q_r(s', a')]$ is lower semicontinuous and $\mathbb{E}_{s' \sim \hat{p}_{s,a}}|V_r^\pi(s')| < \infty$. For any $\pi$ and $Q_c(s', a')$ in (6), $V_c^\pi(s') = \mathbb{E}_{a' \sim \pi(\cdot|s')}[Q_c(s', a')]$ is upper semicontinuous and $\mathbb{E}_{s' \sim \hat{p}_{s,a}}|V_c^\pi(s')| < \infty$.*

**Assumption 2.** *Optimal transport plans exist for the worst-case optimization problems over transition models in (5) and (6).*

Note that Assumptions 1–2 correspond to assumptions in Blanchet & Murthy (2019) applied to our setting. In practice, the use of neural network representations results in continuous value functions, which are bounded for the common case when rewards and costs are bounded, respectively. A sufficient condition for Assumption 2 to hold is if $\mathcal{S}$ is compact, or if we restrict our attention to a compact subset of next states in our definition of $\mathcal{P}_{s,a}$ which is reasonable in practice. Blanchet & Murthy (2019) also provide other sufficient conditions for Assumption 2 to hold.

Under these assumptions, we can reformulate the robust Bellman operators in (5) and (6) to arrive at a tractable result that can be efficiently implemented in a deep RL setting.

**Theorem 1.** *Let Assumptions 1–2 hold, and let $\mathcal{G}$ be the set of all functions from $\mathcal{S}$ to $\mathcal{S}$. Then, we have*

$$\mathcal{T}^\pi_{\mathcal{P},r} Q_r(s,a) = r(s,a) + \gamma \mathop{\mathbb{E}}_{\hat{s}' \sim \hat{p}_{s,a}} \left[ V^\pi_r(g^r_{s,a}(\hat{s}')) \right], \tag{8}$$

$$\mathcal{T}^\pi_{\mathcal{P},c} Q_c(s,a) = c(s,a) + \gamma \mathop{\mathbb{E}}_{\hat{s}' \sim \hat{p}_{s,a}} \left[ V^\pi_c(g^c_{s,a}(\hat{s}')) \right], \tag{9}$$

*where for a given state-action pair $(s,a) \in \mathcal{S} \times \mathcal{A}$ we have*

$$g^r_{s,a} \in \arg \min_{g \in \mathcal{G}} \mathop{\mathbb{E}}_{\hat{s}' \sim \hat{p}_{s,a}} \left[ V^\pi_r(g(\hat{s}')) \right] \quad \text{s.t.} \quad \mathop{\mathbb{E}}_{\hat{s}' \sim \hat{p}_{s,a}} \left[ d_{s,a}(\hat{s}', g(\hat{s}')) \right] \leq \epsilon_{s,a}, \tag{10}$$

$$g^c_{s,a} \in \arg \max_{g \in \mathcal{G}} \mathop{\mathbb{E}}_{\hat{s}' \sim \hat{p}_{s,a}} \left[ V^\pi_c(g(\hat{s}')) \right] \quad \text{s.t.} \quad \mathop{\mathbb{E}}_{\hat{s}' \sim \hat{p}_{s,a}} \left[ d_{s,a}(\hat{s}', g(\hat{s}')) \right] \leq \epsilon_{s,a}. \tag{11}$$

*Proof.* First, we leverage results from Blanchet & Murthy (2019) to show that optimal transport strong duality holds for the worst-case optimization problems over transition models in (5) and (6) under Assumption 1. Next, we establish that the optimization problems in (5) and (6) share the same dual problems as (10) and (11), respectively. Finally, we use Assumption 2 to show that strong duality also holds for (10) and (11), which proves the result. See the Appendix for details. $\qquad\square$

For the optimization problems in (5) and (6), note that the optimal transport plan $\nu^*$ from the nominal transition model $\hat{p}_{s,a}$ to the *worst-case* transition model in $\mathcal{P}_{s,a}$ can be characterized by a deterministic perturbation function in state space (Blanchet & Murthy, 2019). Utilizing this structure, Theorem 1 demonstrates that we can calculate the robust Bellman operators $\mathcal{T}^\pi_{\mathcal{P},r}$ and $\mathcal{T}^\pi_{\mathcal{P},c}$ by using samples collected from $\hat{p}_{s,a}$, and adversarially perturbing the next state samples according to (10) and (11), respectively. We refer to the resulting changes in state transitions as *Optimal Transport Perturbations (OTP)*. As a result, we have replaced difficult optimization problems over transition models in (5) and (6) with the tractable problems of computing Optimal Transport Perturbations in state space. See Figure 1 for an illustration. Theorem 1 represents the main theoretical contribution of our work, which directly motivates an efficient deep RL implementation of robust and safe RL.

Finally, note that these perturbed state transitions are only needed to calculate the Bellman targets in (8) and (9), which we use to train the robust Q functions $Q^\pi_{\mathcal{P},r}(s,a)$ and $Q^\pi_{\mathcal{P},c}(s,a)$. Therefore, unlike other adversarial approaches to robust RL (Pinto et al., 2017; Tessler et al., 2019a; Vinitsky et al., 2020), our Optimal Transport Perturbations have no impact on trajectory rollouts in the nominal training environment. Instead, these perturbations are applied *after* data collection to construct worst-case *virtual* state transitions.

## 6 Perturbation networks for deep reinforcement learning

From Theorem 1, we can calculate Bellman targets for our robust Q functions $Q^\pi_{\mathcal{P},r}(s,a)$ and $Q^\pi_{\mathcal{P},c}(s,a)$ by considering adversarially perturbed versions of next states sampled from $\hat{p}_{s,a}$. We can construct these adversarial perturbations by solving (10) and (11), respectively. Note that we can combine the perturbation functions $g^r_{s,a}, g^c_{s,a} \in \mathcal{G}$ from Theorem 1 across state-action pairs by including $(s,a)$ as input for context, in addition to the next state $\hat{s}'$ to be perturbed. This leads to the combined perturbation functions $g^r, g^c : \mathcal{S} \times \mathcal{A} \times \mathcal{S} \to \mathcal{S}$, where $g^r(s,a,\hat{s}') = g^r_{s,a}(\hat{s}')$ and $g^c(s,a,\hat{s}') = g^c_{s,a}(\hat{s}')$. We let $\mathcal{F}$ be the set of all functions from $\mathcal{S} \times \mathcal{A} \times \mathcal{S}$ to $\mathcal{S}$, with $g^r, g^c \in \mathcal{F}$.

In the context of deep RL, we consider a class of perturbation functions $\mathcal{F}_\delta \subseteq \mathcal{F}$ parameterized by a neural network $\delta : \mathcal{S} \times \mathcal{A} \times \mathcal{S} \to \mathcal{S}$. In our experiments, we consider tasks where $\mathcal{S} = \mathbb{R}^n$ and we apply multiplicative perturbations to state transitions. In particular, we consider perturbation functions of the form

$$g_\delta(s,a,\hat{s}') = s + (\hat{s}' - s)(1 + \delta(s,a,\hat{s}')), \tag{12}$$

where $\delta(s,a,\hat{s}') \in \mathbb{R}^n$ and all operations are performed per-coordinate. By defining $\mathcal{F}_\delta$ in this way, we obtain plausible adversarial transitions that are interpretable, where $\delta(s,a,\hat{s}')$ represents the percentage change to the nominal state transition in each coordinate.

---

**Algorithm 1:** Safe RL with Optimal Transport Perturbations

---

**Input:** initial policy $\pi_0$; critics $Q_{\theta_r}, Q_{\theta_c}$; OTP networks $\delta_r, \delta_c$.

**for** $k = 0, 1, 2, \ldots$ **do**

    Collect data $\tau \sim (\pi_k, \hat{p})$ and store it in $\mathcal{D}$.

    **for** *K updates* **do**

        Sample a batch of data $(s, a, r, c, \hat{s}') \sim \mathcal{D}$.

        Update OTP networks $\delta_r, \delta_c$ according to (13) and (14).

        Calculate Bellman targets in (15) and (16), and update critics $Q_{\theta_r}, Q_{\theta_c}$ to minimize
        $\mathcal{L}_{\mathcal{P},r}(\theta_r), \mathcal{L}_{\mathcal{P},c}(\theta_c)$.

        Update policy $\pi$ according to (4).

    **end**

**end**

---

Using $d_{s,a}$ from (7), we have that

$$d_{s,a}(\hat{s}', g_\delta(s, a, \hat{s}')) = \frac{1}{n} \|\delta(s, a, \hat{s}')\|_2^2.$$

Then, following common practice in off-policy deep RL, we combine the perturbation function updates in (10) and (11) across state-action pairs by averaging over samples from the replay buffer. By doing so, we can efficiently update our perturbation networks in a deep RL setting according to

$$\delta_r \in \arg\min_\delta \; \mathbb{E}_{(s,a,\hat{s}') \sim \mathcal{D}} \left[ V_r^\pi(g_\delta(s, a, \hat{s}')) \right] \quad \text{s.t.} \quad \mathbb{E}_{(s,a,\hat{s}') \sim \mathcal{D}} \left[ \|\delta(s, a, \hat{s}')\|_2^2 \right] \leq n\epsilon_\delta^2, \tag{13}$$

$$\delta_c \in \arg\max_\delta \; \mathbb{E}_{(s,a,\hat{s}') \sim \mathcal{D}} \left[ V_c^\pi(g_\delta(s, a, \hat{s}')) \right] \quad \text{s.t.} \quad \mathbb{E}_{(s,a,\hat{s}') \sim \mathcal{D}} \left[ \|\delta(s, a, \hat{s}')\|_2^2 \right] \leq n\epsilon_\delta^2, \tag{14}$$

where $(s, a, \hat{s}') \sim \mathcal{D}$ are transitions collected in the nominal environment and $\epsilon_\delta$ represents the average per-coordinate magnitude of $\delta(s, a, \hat{s}')$ with $\mathbb{E}_{(s,a) \sim \mathcal{D}} [\epsilon_{s,a}] = \epsilon_\delta^2$. In practice, we apply gradient-based updates to the Lagrangian relaxations of (13) and (14), and we update the corresponding dual variables throughout training. It is also possible to satisfy perturbation function constraints at every state-action pair through the use of clipping mechanisms, if desired. Note that any violation of the perturbation function constraints in practice will only lead to additional robustness and will not negatively impact safety.

We train separate reward and cost perturbation networks $\delta_r$ and $\delta_c$, and we apply the resulting Optimal Transport Perturbations to calculate the Bellman targets in (8) and (9) for training the robust Q functions $Q_{\mathcal{P},r}^\pi(s, a)$ and $Q_{\mathcal{P},c}^\pi(s, a)$. For $(s, a, \hat{s}') \sim \mathcal{D}$, we consider the sample-based estimates

$$\hat{\mathcal{T}}_{\mathcal{P},r}^\pi Q_r(s, a) = r(s, a) + \gamma V_r^\pi(g_{\delta_r}(s, a, \hat{s}')), \tag{15}$$

$$\hat{\mathcal{T}}_{\mathcal{P},c}^\pi Q_c(s, a) = c(s, a) + \gamma V_c^\pi(g_{\delta_c}(s, a, \hat{s}')). \tag{16}$$

## 7 Algorithm

We summarize our approach to robust and safe RL in Algorithm 1. At every update, we sample previously collected data from a replay buffer $\mathcal{D}$. We update our reward and cost perturbation networks $\delta_r$ and $\delta_c$ according to (13) and (14), respectively. Then, we estimate Bellman targets according to (15) and (16), which we use to update our critics via standard temporal difference learning loss functions. We consider

Table 1: Aggregate performance summary

| Algorithm | % Safe[†] | Normalized Ave.[‡] | | Rollouts Require[§] | |
|---|---|---|---|---|---|
| | | Reward | Cost | Adversary | Simulator |
| Safe RL | 51%[*] | 1.00[*] | 1.00[*] | No | No |
| **OTP** | **87%** | **1.06** | **0.34** | **No** | **No** |
| Adversarial RL (10%) | 88% | 0.95[*] | 0.28 | Yes | No |
| Adversarial RL (5%) | 82%[*] | 1.05 | 0.48[*] | Yes | No |
| Domain Randomization | 76%[*] | 1.14[*] | 0.72[*] | No | Yes |
| Domain Randomization (OOD) | 55%[*] | 1.02 | 1.02[*] | No | Yes |
| Risk-Averse Model Uncertainty | 80%[*] | 1.08 | 0.51[*] | No | No |

[†] Percentage of policies that satisfy the safety constraint across all tasks and test environments.

[‡] Normalized relative to the average performance of standard safe RL for each task and test environment.

[§] Denotes need for adversary or simulator during data collection (i.e., trajectory rollouts) for training.

[*] Statistically significant difference ($p < 0.05$) compared to OTP using a paired t-test.

parameterized critics $Q_{\theta_r}$ and $Q_{\theta_c}$, and we optimize their parameters to minimize the loss functions

$$\mathcal{L}_{\mathcal{P},r}(\theta_r) = \mathop{\mathbb{E}}_{(s,a,\hat{s}')\sim\mathcal{D}}\left[\left(Q_{\theta_r}(s,a) - \hat{\mathcal{T}}_{\mathcal{P},r}^{\pi}\bar{Q}_{\theta_r}(s,a)\right)^2\right],$$

$$\mathcal{L}_{\mathcal{P},c}(\theta_c) = \mathop{\mathbb{E}}_{(s,a,\hat{s}')\sim\mathcal{D}}\left[\left(Q_{\theta_c}(s,a) - \hat{\mathcal{T}}_{\mathcal{P},c}^{\pi}\bar{Q}_{\theta_c}(s,a)\right)^2\right],$$

where $\bar{Q}_{\theta_r}$ and $\bar{Q}_{\theta_c}$ represent target critic networks. Finally, we use these critic estimates to update our policy according to (4).

Compared to standard safe RL methods, the only additional components of our approach are the perturbation networks used to apply Optimal Transport Perturbations, which we train alongside the critics and the policy using standard gradient-based methods. Otherwise, the computations for updating the critics and policy remain unchanged. Therefore, it is simple to incorporate our OTP framework into existing deep safe RL algorithms in order to provide robustness guarantees on performance and safety.

## 8 Experiments

We analyze the use of Optimal Transport Perturbations for robust and safe RL on continuous control tasks with safety constraints in the Real-World RL Suite (Dulac-Arnold et al., 2020; 2021). We follow the same experimental design used in Queeney & Benosman (2023). In particular, we consider 5 constrained tasks over 3 domains (Cartpole Swingup, Walker Walk, Walker Run, Quadruped Walk, and Quadruped Run), which all have horizons of 1,000 with $r(s,a) \in [0,1]$ and $c(s,a) \in \{0,1\}$. In all tasks, we consider a safety budget of $B = 100$. We train policies in a nominal training environment for 1 million steps over 5 random seeds, and we evaluate the robustness of the learned policies in terms of both performance and safety across a range of perturbed test environments. See the Appendix for details on the safety constraints and environment perturbations considered for each task.

Our Optimal Transport Perturbations can be combined with many popular safe RL algorithms that perform policy updates according to (2) by replacing standard Q functions with robust Q functions as in (4), which is a benefit of our methodology. In our experiments, we consider the safe RL algorithm Constraint-Rectified Policy Optimization (CRPO) (Xu et al., 2021), and we use the unconstrained deep RL algorithm Maximum a Posteriori Policy Optimization (MPO) (Abdolmaleki et al., 2018) to calculate policy updates in CRPO. For a fair comparison, we apply CRPO with MPO policy updates as the baseline safe RL algorithm in every method we consider in our experiments. See the Appendix for additional results using a Lagrangian-based safe RL update method (Tessler et al., 2019b; Ray et al., 2019). We train a multivariate Gaussian policy,

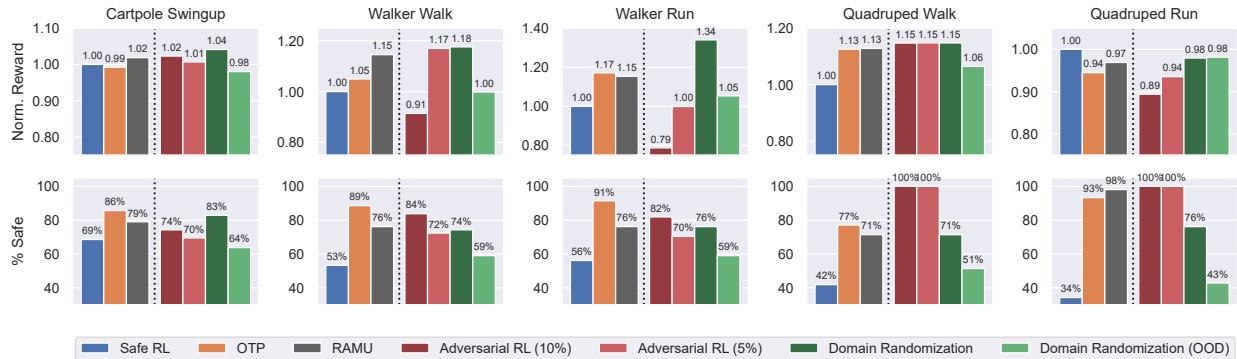

Figure 3: Performance summary by task, aggregated across test environments. Performance of adversarial RL is evaluated without adversarial interventions. Algorithms to the right of the dotted lines require additional assumptions during data collection compared to standard safe RL (see Table 1 for details). Top: Average total reward, normalized relative to the average performance of standard safe RL for each test environment. Bottom: Percentage of policies that satisfy the safety constraint across all test environments.

where the mean and diagonal covariance at a given state are parameterized by a neural network. We also consider separate neural network parameterizations for the reward and cost critics. See the Appendix for additional implementation details, including network architectures and values of all hyperparameters.[1]

We incorporate robustness into the baseline safe RL algorithm in four ways: (i) Optimal Transport Perturbations, (ii) adversarial RL using the action-robust PR-MDP framework from Tessler et al. (2019a) applied to the safety constraint, (iii) the soft-robust approach of domain randomization (Peng et al., 2018), and (iv) the distributionally robust approach of Risk-Averse Model Uncertainty (RAMU) from Queeney & Benosman (2023). For our Optimal Transport Perturbations, we consider $\epsilon_\delta = 0.02$ (i.e., 2% perturbations on average). Figure 3 and Table 1 summarize the performance of all algorithms at deployment time, where performance metrics are aggregated across the range of perturbed test environments. See the Appendix for detailed results across all test environments. Note that the training process of each algorithm only considers safety for a specific set of environments, so we do not expect the resulting policies to remain safe across all possible test cases.

## 8.1 Comparison to safe reinforcement learning

Figure 4 compares our OTP framework to standard safe RL across all tasks and perturbed test environments. By applying Optimal Transport Perturbations to the objective and constraint in safe RL, we achieve meaningful test-time improvements compared to the standard non-robust version of safe RL. While in most cases we observe a decrease in total rewards in the nominal environment in order to achieve robustness, as expected, on average our framework leads to an increase in total rewards of 1.06x relative to safe RL across the range of test environments. Most importantly, we see a significant improvement in safety, with our algorithm satisfying constraints in 87% of test cases (compared to 51% for safe RL) and incurring 0.34x the costs of safe RL, on average. Note that we achieve this robustness while collecting data from the same training environment considered in standard safe RL, without requiring adversarial interventions in the environment or domain knowledge on the structure of the perturbed test environments.

## 8.2 Comparison to adversarial reinforcement learning

Next, we compare our approach to the PR-MDP framework (Tessler et al., 2019a), an adversarial RL method that randomly applies adversarial actions a percentage of the time during training. In order to apply this method to the safe RL setting, we train the adversary to maximize costs. We apply the default probability of

---

[1]Code is publicly available at `https://github.com/jqueeney/robust-safe-rl`.

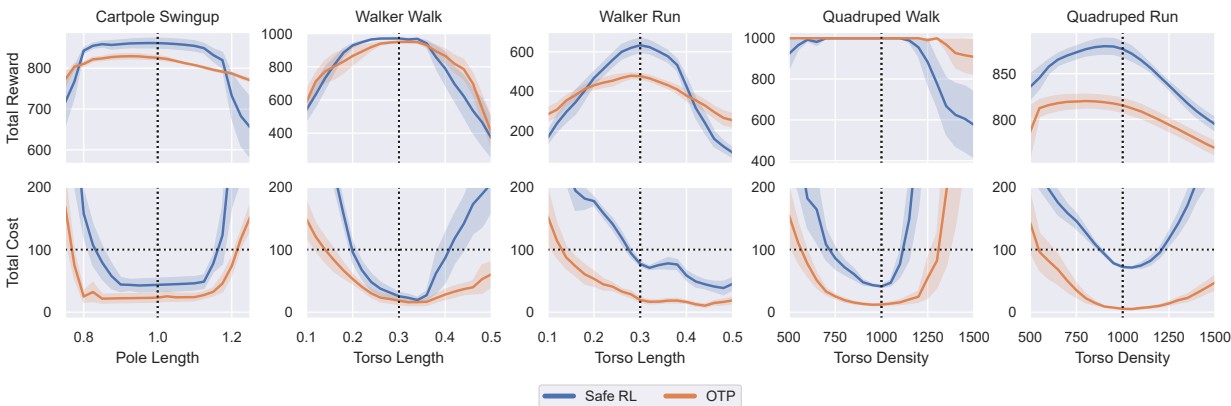

Figure 4: Comparison of OTP with standard safe RL across tasks and test environments. Shading denotes one standard error across policies. Vertical dotted lines represent nominal training environment. Top: Total reward. Bottom: Total cost, where horizontal dotted lines represent the safety budget and values below these lines represent safety constraint satisfaction.

intervention of 10% considered in Tessler et al. (2019a). As shown in Figure 3, this adversarial approach leads to robust constraint satisfaction at test time in the Quadruped tasks. Our OTP framework, on the other hand, leads to improved constraint satisfaction in the remaining 3 out of 5 tasks. However, the robust safety demonstrated by adversarial RL also results in lower total rewards on average, and is the only robust approach in Table 1 that underperforms standard safe RL in terms of reward. In order to improve performance with respect to reward, we also considered adversarial RL with a lower probability of intervention of 5%. While this less adversarial implementation is comparable to our OTP framework in terms of total rewards, it leads to a decrease in safety constraint satisfaction to 82%. Therefore, our OTP formulation demonstrates the safety benefits of the more adversarial setting and the reward benefits of the less adversarial setting.

In addition, an important drawback of adversarial RL is that it requires the intervention of an adversary in the training environment. Therefore, in order to achieve robust safety at deployment time, adversarial RL incurs additional cost during training due to the presence of an adversary. Even in the Quadruped tasks where adversarial RL results in near-zero cost at deployment time, Figure 5 shows that adversarial RL leads to the highest total cost during training due to adversarial interventions. In many real-world situations, this additional cost during training is undesirable. Our OTP framework, on the other hand, achieves the lowest total cost during training, while also resulting in robust safety when deployed in perturbed environments. This is due to the fact that our Optimal Transport Perturbations do not impact the data collection process during training.

## 8.3 Comparison to domain randomization

We now compare our OTP framework to the soft-robust approach of domain randomization (Peng et al., 2018), which assumes access to a range of environments during training through the use of a simulator. By training across half of the test environments, domain randomization achieves strong performance across test cases in terms of reward (1.14x compared to safe RL, on average), which was the motivation for its development in the setting of sim-to-real transfer. However, domain randomization only satisfies safety constraints in 76% of test cases, which is lower than both OTP and adversarial RL which explicitly consider robust formulations. This is likely due to its soft-robust approach that focuses on average performance across the training distribution.

It is important to note that domain randomization not only requires access to a range of training environments, it also requires prior knowledge on the structure of potential disturbances to define its training distribution. In order to evaluate the case where we lack domain knowledge, we consider an out-of-distribution (OOD) version of domain randomization that is trained on a distribution over a different parameter than the one varied at

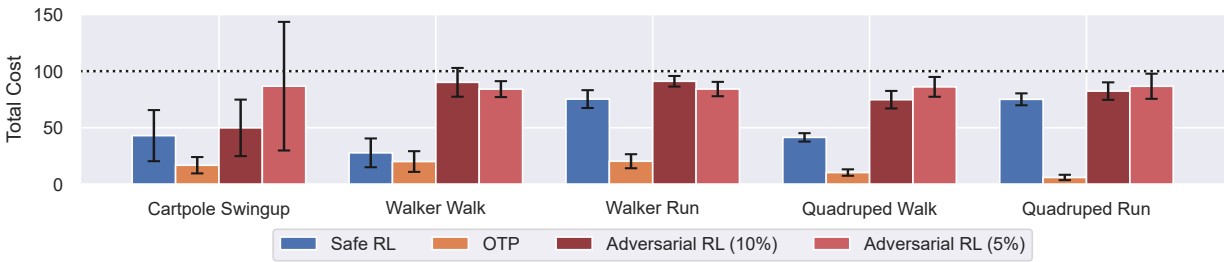

Figure 5: Average final training cost in the nominal training environment. Training cost of adversarial RL includes impact of adversarial interventions. Black bars denote one standard deviation across policies. Horizontal dotted line represents safety budget.

test time. When the training distribution is not appropriately selected, we see that domain randomization provides little benefit compared to standard safe RL. Our OTP framework, on the other hand, guarantees robust and safe performance under general forms of environment uncertainty while only collecting data from a single training environment.

### 8.4 Comparison to distributionally robust safe reinforcement learning

Finally, we compare our robust and safe RL method to a distributionally robust safe RL algorithm from Queeney & Benosman (2023) called Risk-Averse Model Uncertainty (RAMU). Distributionally robust RL (Xu & Mannor, 2010; Yu & Xu, 2016) considers a distribution over transition models $p \sim \mu$, and provides robustness to worst-case *distributions over transition models* in an ambiguity set centered around $\mu$. RAMU applies a coherent distortion risk measure over $\mu$ to achieve distributionally robust guarantees in a safe RL setting. For specific choices of $\mu$, this approach can be implemented without requiring detailed simulator access. Because distributionally robust RL provides robustness to worst-case *distributions over transition models* instead of worst-case transition models, RAMU results in a softer notion of robustness compared to the robust and safe RL problem in (3).

We see in Figure 3 and Table 1 that RAMU slightly improves performance in terms of total rewards compared to our OTP framework due to its less conservative notion of robustness, but this comes at the expense of safety constraint satisfaction. The stronger robustness guarantees of our problem formulation translate to a statistically significant improvement in safety at deployment time compared to distributionally robust safe RL. Our OTP framework remains safe in 87% of test environments, compared to 80% for RAMU. Overall, these experimental results are in line with the different robustness guarantees associated with these frameworks.

## 9 Conclusion and broader impacts

In this work, we have developed a general, efficient framework for robust and safe RL. Through the use of optimal transport theory, we demonstrated that we can guarantee robustness to general forms of environment disturbances by applying adversarial perturbations to observed state transitions. These Optimal Transport Perturbations can be efficiently implemented by constructing virtual transitions without impacting data collection during training, and can be easily combined with existing techniques for safe RL to provide protection against unknown disturbances. Because our framework makes limited assumptions on the data collection process during training and does not require directly modifying the environment, it should be compatible with many real-world decision making applications. As a result, we hope that our work represents a promising step towards trustworthy deep RL algorithms that can be reliably deployed to improve real-world decision making.

Despite this progress, there are limitations to our approach that should be considered prior to potential deployment. Our use of the RC-MDP framework guarantees robust performance and safety for all environments in the uncertainty set $\mathcal{P}$, which requires specifying the transport cost function $d_{s,a}$ and radius $\epsilon_{s,a}$ to achieve

the desired robustness without being overly conservative. In addition, we consider the C-MDP definition of safety given by a constraint on expected total costs. In order to develop robust control methods for safety-critical scenarios that instead require avoiding sets of unsafe states, the combination of our OTP framework with control-theoretic approaches to safety represents an interesting avenue for future work. Finally, our experimental analysis focused on a set of 5 simulated continuous control tasks with deterministic dynamics from the Real-World RL Suite. Given the flexibility of our framework, we believe there are exciting opportunities to apply our methodology across a variety of additional tasks and problem settings in future work, including complex robotics applications where accurate simulators are not available and real-world data collection is required for training.

**Acknowledgments**

This research was partially supported by the NSF under grants CCF-2200052, CNS-1645681, CNS-2149511, DMS-1664644, ECCS-1931600, and IIS-1914792, by the ONR under grants N00014-19-1-2571 and N00014-21-1-2844, by the NIH under grants R01 GM135930 and UL54 TR004130, by AFOSR under grant FA9550-19-1-0158, by ARPA-E under grant DE-AR0001282, by the MathWorks, and by the Boston University Kilachand Fund for Integrated Life Science and Engineering. The majority of this work was done while James Queeney was with the Division of Systems Engineering, Boston University. He is currently with and exclusively supported by Mitsubishi Electric Research Laboratories.

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

## A Proofs

In order to prove the tractable reformulation in Theorem 1, we will make use of the following result.

**Lemma 1.** *Let Assumption 1 hold. Then, we have*

$$\mathcal{T}_{\mathcal{P},r}^\pi Q_r(s,a) = r(s,a) + \gamma \sup_{\lambda \geq 0} \mathbb{E}_{\hat{s}' \sim \hat{p}_{s,a}} \left[ \inf_{s' \in \mathcal{S}} V_r^\pi(s') + \lambda \left( d_{s,a}(\hat{s}', s') - \epsilon_{s,a} \right) \right], \tag{17}$$

$$\mathcal{T}_{\mathcal{P},c}^\pi Q_c(s,a) = c(s,a) + \gamma \inf_{\lambda \geq 0} \mathbb{E}_{\hat{s}' \sim \hat{p}_{s,a}} \left[ \sup_{s' \in \mathcal{S}} V_c^\pi(s') - \lambda \left( d_{s,a}(\hat{s}', s') - \epsilon_{s,a} \right) \right]. \tag{18}$$

Note that

$$\inf_{p_{s,a} \in \mathcal{P}_{s,a}} \mathbb{E}_{s' \sim p_{s,a}} [V_r^\pi(s')] = - \sup_{p_{s,a} \in \mathcal{P}_{s,a}} \mathbb{E}_{s' \sim p_{s,a}} [-V_r^\pi(s')].$$

Therefore, in this section we only prove results for the robust cost Bellman operator $\mathcal{T}_{\mathcal{P},c}^\pi$. Results related to the robust reward Bellman operator $\mathcal{T}_{\mathcal{P},r}^\pi$ follow immediately by applying the same proofs after an appropriate change in signs.

### A.1 Proof of Lemma 1

*Proof.* Under Assumption 1, note that Assumption (A1) and Assumption (A2) of Blanchet & Murthy (2019) are satisfied for the worst-case optimization problem over transition models in (6). Assumption (A1) is satisfied by our definition of optimal transport cost, and Assumption (A2) is satisfied by our Assumption 1.

Then, according to Theorem 1 in Blanchet & Murthy (2019), optimal transport strong duality holds for the optimization problem in (6). Therefore, we have that

$$\sup_{p_{s,a} \in \mathcal{P}_{s,a}} \mathbb{E}_{s' \sim p_{s,a}} [V_c^\pi(s')] = \inf_{\lambda \geq 0} \mathbb{E}_{\hat{s}' \sim \hat{p}_{s,a}} \left[ \sup_{s' \in \mathcal{S}} V_c^\pi(s') - \lambda \left( d_{s,a}(\hat{s}', s') - \epsilon_{s,a} \right) \right].$$

By substituting this result into (6), we arrive at the result in (18). $\qquad\square$

### A.2  Proof of Theorem 1

*Proof.* First, we show that (11) and the worst-case optimization problem over transition models in (6) share the same dual problem. We write the dual problem to (11) as

$$\inf_{\lambda \geq 0} \sup_{g \in \mathcal{G}} \mathbb{E}_{\hat{s}' \sim \hat{p}_{s,a}} [V_c^\pi(g(\hat{s}'))] - \lambda \left( \mathbb{E}_{\hat{s}' \sim \hat{p}_{s,a}} [d_{s,a}(\hat{s}', g(\hat{s}'))] - \epsilon_{s,a} \right)$$
$$= \inf_{\lambda \geq 0} \sup_{g \in \mathcal{G}} \mathbb{E}_{\hat{s}' \sim \hat{p}_{s,a}} [V_c^\pi(g(\hat{s}')) - \lambda \left( d_{s,a}(\hat{s}', g(\hat{s}')) - \epsilon_{s,a} \right)].$$

Using the definition of $\mathcal{G}$, we can rewrite this as

$$\inf_{\lambda \geq 0} \mathbb{E}_{\hat{s}' \sim \hat{p}_{s,a}} \left[ \sup_{s' \in \mathcal{S}} V_c^\pi(s') - \lambda \left( d_{s,a}(\hat{s}', s') - \epsilon_{s,a} \right) \right], \tag{19}$$

which appears in the right-hand side of (18) from Lemma 1. As shown in Lemma 1, (19) is also the dual to the optimization problem in (6), and optimal transport strong duality holds.

Next, we show that strong duality holds between (11) and (19). Let $\lambda^*$ be the optimal dual variable in (19), and let

$$g_{s,a}^*(\hat{s}') \in \arg \max_{s' \in \mathcal{S}} V_c^\pi(s') - \lambda^* \left( d_{s,a}(\hat{s}', s') - \epsilon_{s,a} \right).$$

We have that $\lambda^*$ and $g_{s,a}^*(\hat{s}')$ exist according to Theorem 1(b) in Blanchet & Murthy (2019) along with Assumption 2, and $g_{s,a}^*$ characterizes the optimal transport plan $\nu^*$ that moves the probability of $\hat{s}'$ under $\hat{p}_{s,a}$ to $g_{s,a}^*(\hat{s}')$. By the complementary slackness results of Theorem 1(b) in Blanchet & Murthy (2019), we also have that

$$\lambda^* \left( \mathbb{E}_{\hat{s}' \sim \hat{p}_{s,a}} \left[ d_{s,a}(\hat{s}', g_{s,a}^*(\hat{s}')) \right] - \epsilon_{s,a} \right) = 0.$$

Therefore,

$$\inf_{\lambda \geq 0} \mathbb{E}_{\hat{s}' \sim \hat{p}_{s,a}} \left[ \sup_{s' \in \mathcal{S}} V_c^\pi(s') - \lambda \left( d_{s,a}(\hat{s}', s') - \epsilon_{s,a} \right) \right] = \mathbb{E}_{\hat{s}' \sim \hat{p}_{s,a}} \left[ V_c^\pi(g_{s,a}^*(\hat{s}')) - \lambda^* \left( d_{s,a}(\hat{s}', g_{s,a}^*(\hat{s}')) - \epsilon_{s,a} \right) \right]$$
$$= \mathbb{E}_{\hat{s}' \sim \hat{p}_{s,a}} \left[ V_c^\pi(g_{s,a}^*(\hat{s}')) \right].$$

Moreover, by the primal feasibility of the optimal transport plan $\nu^*$, we have that

$$\mathbb{E}_{\hat{s}' \sim \hat{p}_{s,a}} \left[ d_{s,a}(\hat{s}', g_{s,a}^*(\hat{s}')) \right] \leq \epsilon_{s,a},$$

so $g_{s,a}^*$ is a feasible solution to (11) with the same objective value as the value of (19). Therefore, strong duality holds between (11) and (19), and $g_{s,a}^*$ is an optimal solution to (11) (i.e., an optimal solution to (11) exists). Then, for any optimal solution $g_{s,a}^c$ to (11), we have that

$$\mathbb{E}_{\hat{s}' \sim \hat{p}_{s,a}} \left[ V_c^\pi(g_{s,a}^c(\hat{s}')) \right] = \mathbb{E}_{\hat{s}' \sim \hat{p}_{s,a}} \left[ V_c^\pi(g_{s,a}^*(\hat{s}')) \right],$$

and the right-hand side of (18) is equivalent to the right-hand side of (9). $\qquad\square$

Table 2: Safety constraints for all tasks

| Task | Safety Constraint | Safety Coefficient |
|---|---|---|
| Cartpole Swingup | Slider Position | 0.30 |
| Walker Walk | Joint Velocity | 0.25 |
| Walker Run | Joint Velocity | 0.30 |
| Quadruped Walk | Joint Angle | 0.15 |
| Quadruped Run | Joint Angle | 0.30 |

Table 3: Perturbation ranges for test environments across domains

| Domain | Perturbation Parameter | Nominal Value | Test Range |
|---|---|---|---|
| Cartpole | Pole Length | 1.00 | $[0.75, 1.25]$ |
| Walker | Torso Length | 0.30 | $[0.10, 0.50]$ |
| Quadruped | Torso Density | 1,000 | $[500, 1,500]$ |

## B Implementation details

### B.1 Safety constraints and environment perturbations

We consider the same experimental design used in Queeney & Benosman (2023) to define our training and test environments. For each task, we consider a single safety constraint defined in the Real-World RL Suite, which we summarize in Table 2. A policy incurs cost in the Cartpole domain when the slider moves outside of a specified range, in the Walker domain when the velocity of any joint exceeds a threshold, and in the Quadruped domain when joint angles are outside of an acceptable range. The specific ranges that result in cost violations are determined by the safety coefficients in Table 2, which can take values in $[0, 1]$ where lower values make cost violations more likely. See Dulac-Arnold et al. (2021) for detailed definitions of the safety constraints we consider.

We evaluate the performance of learned policies across a range of test environments different from the training environment. We define these test environments by varying a simulator parameter in each domain across a range of values, which are listed in Table 3. We vary the length of the pole in the Cartpole domain, the length of the torso in the Walker domain, and the density of the torso in the Quadruped domain. Note that the parameter value associated with the nominal training environment is in the center of the range of parameter values considered at test time. For each test environment, we evaluate performance of the learned policies via 10 trajectory rollouts.

Finally, note that the domain randomization baselines consider a range of environments during training. We consider the same training distributions for domain randomization as in Queeney & Benosman (2023). As summarized in Table 4, in-distribution domain randomization applies a uniform distribution over the middle 50% of the parameter values considered at test time. For the out-of-distribution variant of domain randomization, we instead consider a uniform distribution over a range of values for a different simulator parameter than the one varied at test time. See Table 4 for details.

### B.2 Network architectures

In our experiments, we consider neural network representations of the policy and critics. We consider networks with 3 hidden layers of 256 units and ELU activations, and we apply layer normalization followed by a tanh activation after the first layer as in Abdolmaleki et al. (2020). We represent the policy as a multivariate Gaussian distribution with diagonal covariance, where at a given state the policy network outputs the mean

Table 4: Perturbation parameters and ranges for domain randomization across domains

| Domain | Perturbation Parameter | Nominal Value | Training Range |
|---|---|---|---|
| In-Distribution | | | |
| Cartpole | Pole Length | 1.00 | $[0.875, 1.125]$ |
| Walker | Torso Length | 0.30 | $[0.20, 0.40]$ |
| Quadruped | Torso Density | 1,000 | $[750, 1,250]$ |
| Out-of-Distribution | | | |
| Cartpole | Pole Mass | 0.10 | $[0.05, 0.15]$ |
| Walker | Contact Friction | 0.70 | $[0.40, 1.00]$ |
| Quadruped | Contact Friction | 1.50 | $[1.00, 2.00]$ |

$\mu(s)$ and diagonal covariance $\Sigma(s)$ of the action distribution. The diagonal of $\Sigma(s)$ is calculated by applying the softplus operator to the outputs of the neural network corresponding to the covariance. In addition to the policy network, we consider separate networks for the reward and cost critics. We maintain target versions of the policy and critic networks using an exponential moving average of the weights with $\tau = 5\mathrm{e}{-3}$. Finally, we also consider neural networks for our perturbation networks $\delta_r$ and $\delta_c$. In this work, we consider small networks with 2 hidden layers of 64 units and ELU activations. We clip the outputs in the range $[-2\epsilon_\delta, 2\epsilon_\delta]$ for additional stability.

### B.3 Algorithm hyperparameters

All of the experimental results in Section 8 build upon the baseline safe RL algorithm CRPO (Xu et al., 2021). At every update, CRPO calculates the current value of the safety constraint based on a batch of sampled data. If the safety constraint is satisfied for the current batch, it applies a policy update to maximize rewards. Otherwise, it applies a policy update to minimize costs. In both cases, we use the unconstrained RL algorithm MPO (Abdolmaleki et al., 2018) to calculate policy updates. MPO calculates a non-parametric target policy with KL divergence $\epsilon_{\mathrm{KL}}$ from the current policy, and updates the current policy towards this target while constraining separate KL divergence contributions from the mean and covariance by $\beta_\mu$ and $\beta_\Sigma$, respectively. We apply per-dimension KL divergence constraints and action penalization using the multi-objective MPO framework (Abdolmaleki et al., 2020) as in Hoffman et al. (2020), and we consider closed-form updates of the temperature parameter used in the non-parametric target policy as in Liu et al. (2022) to account for the immediate switching between objectives in CRPO. See Table 5 for all important hyperparameter values associated with the implementation of policy updates using MPO, and see Abdolmaleki et al. (2018) for additional details.

For our OTP framework, we update the perturbation networks alongside the policy and critics. We combine the perturbation function updates in (10) and (11) across state-action pairs by averaging over samples from the replay buffer, and we apply a radius of $\epsilon_\delta^2$. This leads to updates of the form

$$g_{\delta_r} \in \arg\min_{g_\delta \in \mathcal{F}_\delta} \mathop{\mathbb{E}}_{(s,a,\hat{s}')\sim\mathcal{D}} \left[ V_r^\pi(g_\delta(s,a,\hat{s}')) \right] \quad \text{s.t.} \quad \mathop{\mathbb{E}}_{(s,a,\hat{s}')\sim\mathcal{D}} \left[ d_{s,a}(\hat{s}', g_\delta(s,a,\hat{s}')) \right] \leq \epsilon_\delta^2,$$

$$g_{\delta_c} \in \arg\max_{g_\delta \in \mathcal{F}_\delta} \mathop{\mathbb{E}}_{(s,a,\hat{s}')\sim\mathcal{D}} \left[ V_c^\pi(g_\delta(s,a,\hat{s}')) \right] \quad \text{s.t.} \quad \mathop{\mathbb{E}}_{(s,a,\hat{s}')\sim\mathcal{D}} \left[ d_{s,a}(\hat{s}', g_\delta(s,a,\hat{s}')) \right] \leq \epsilon_\delta^2,$$

where $\mathcal{F}_\delta$ represents the class of perturbation functions with the form in (12). We consider $d_{s,a}$ given by (7) to arrive at the perturbation network updates in (13) and (14), where $\epsilon_\delta$ determines the average per-coordinate magnitude of the outputs of $\delta_r$ and $\delta_c$. We apply gradient-based updates on the Lagrangian relaxations of (13) and (14), and we also update the corresponding dual variables throughout training. See Table 5 for hyperparameter values associated with our OTP framework.

Table 5: Network architectures and algorithm hyperparameters used in experiments

| General | |
|---|---|
| Batch size per update | 256 |
| Updates per environment step | 1 |
| Discount rate ($\gamma$) | 0.99 |
| Target network exponential moving average ($\tau$) | 5e−3 |

| Policy | |
|---|---|
| Layer sizes | 256, 256, 256 |
| Layer activations | ELU |
| Layer norm + tanh on first layer | Yes |
| Initial standard deviation | 0.3 |
| Learning rate | 1e−4 |
| Non-parametric KL ($\epsilon_{\mathrm{KL}}$) | 0.10 |
| Action penalty KL | 1e−3 |
| Action samples per update | 20 |
| Parametric mean KL ($\beta_\mu$) | 0.01 |
| Parametric covariance KL ($\beta_\Sigma$) | 1e−5 |
| Parametric KL dual learning rate | 0.01 |

| Critics | |
|---|---|
| Layer sizes | 256, 256, 256 |
| Layer activations | ELU |
| Layer norm + tanh on first layer | Yes |
| Learning rate | 1e−4 |

| Optimal Transport Perturbations | |
|---|---|
| Layer sizes | 64, 64 |
| Layer activations | ELU |
| Layer norm + tanh on first layer | No |
| Output clipping | $[-2\epsilon_\delta, 2\epsilon_\delta]$ |
| Learning rate | 1e−4 |
| Dual learning rate | 0.01 |
| Per-coordinate perturbation magnitude ($\epsilon_\delta$) | 0.02 |

Finally, we also implement adversarial RL, domain randomization, and RAMU using CRPO with MPO policy updates. The adversarial policy used in the PR-MDP framework is updated using MPO with the goal of maximizing costs. Using the default settings from Tessler et al. (2019a), we apply one adversary update for every 10 policy updates. Domain randomization considers the same updates as the CRPO baseline, but collects data from the range of training environments summarized in Table 4. RAMU applies the distribution over transition models and risk measure used in Queeney & Benosman (2023).

### B.4 Computational resources

All experiments were run on a Linux cluster with 2.9 GHz Intel Gold processors and NVIDIA A40 and A100 GPUs. The Real-World RL Suite (Dulac-Arnold et al., 2020; 2021) is available under the Apache License 2.0. Using code that has not been optimized for execution speed, each combination of algorithm and task

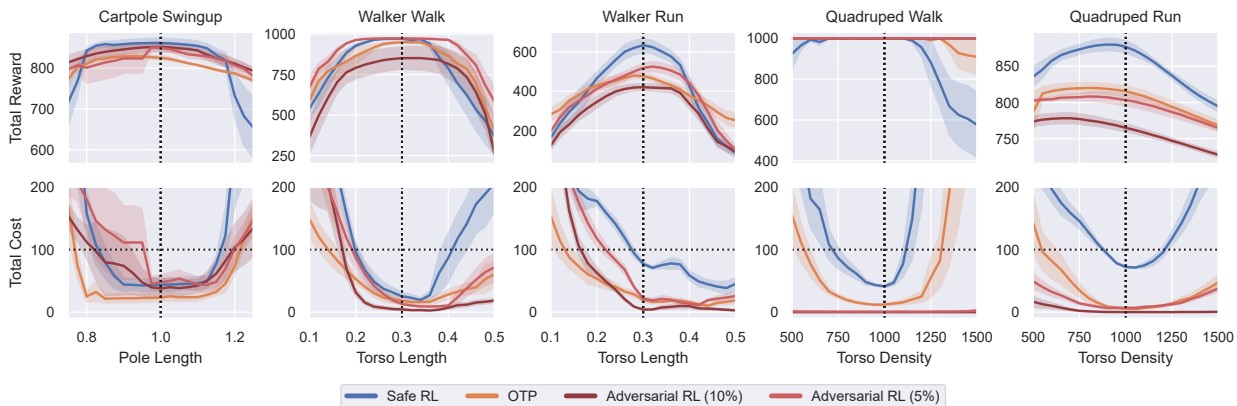

Figure 6: Comparison with adversarial RL. Performance of adversarial RL is evaluated without adversarial interventions. Shading denotes one standard error across policies. Vertical dotted lines represent nominal training environment. Top: Total reward. Bottom: Total cost, where horizontal dotted lines represent the safety budget and values below these lines represent safety constraint satisfaction.

required approximately one day of wall-clock time on a single GPU to train policies for 1 million steps across 5 random seeds.

## C Additional experimental results

### C.1 Detailed experimental results across test environments

In this section, we include detailed results across tasks and test environments for all algorithms listed in Table 1. Figure 6 compares safe RL and OTP to both variations of adversarial RL using different probabilities of adversary intervention. In general, adversarial RL with 5% intervention probability achieves similar total reward to OTP across all tasks, while the more adversarial 10% implementation leads to overly conservative performance. Both versions of adversarial RL generate strong, safe performance in the Quadruped tasks, while our OTP framework results in more consistent constraint satisfaction in the other 3 out of 5 tasks. Importantly, our OTP framework accomplishes this without requiring adversarial interventions during data collection.

Figure 7 shows the performance of domain randomization across tasks and environment perturbations. The grey shaded areas represent the ranges of the training distributions for the in-distribution implementation of domain randomization. We see that domain randomization leads to strong, robust performance in terms of rewards across all test cases, as well as improved constraint satisfaction in perturbed environments compared to standard safe RL with a single training environment. However, in tasks such as Walker Run and Quadruped Run, domain randomization does not robustly satisfy safety constraints for test environments that were not seen during training. This issue is amplified in the case of out-of-distribution domain randomization, which does not demonstrate consistent robustness benefits compared to standard safe RL. In fact, it even leads to an increase in constraint-violating test cases in Cartpole Swingup compared to safe RL. This demonstrates that training on multiple environments does not necessarily lead to robust performance. Instead, domain knowledge is critical in order for domain randomization to work well in practice.

Finally, Figure 8 shows the performance of RAMU across tasks and test environments. The distributionally robust approach of RAMU leads to slight improvements in terms of rewards compared to OTP across most tasks and test environments, while our OTP framework delivers more robust safety constraint satisfaction in 4 out of 5 tasks. This highlights the different robustness guarantees associated with these frameworks.

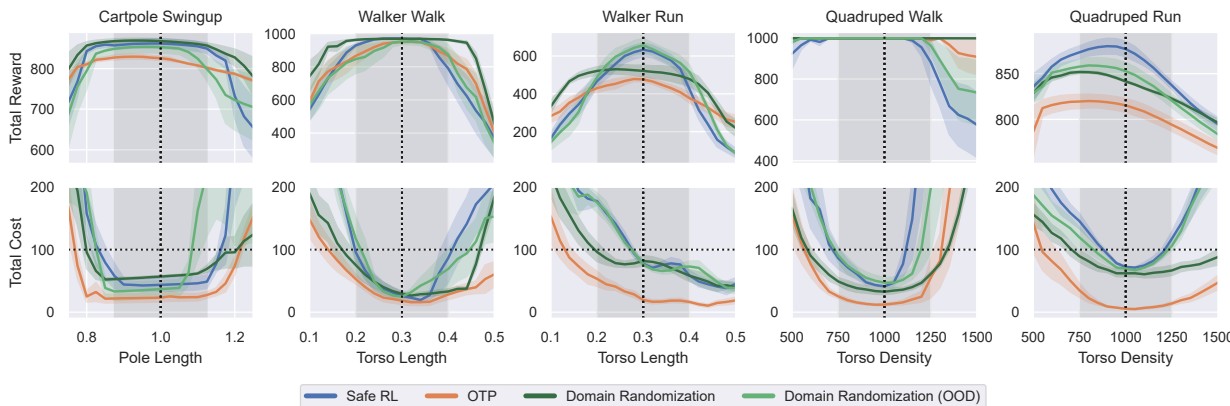

Figure 7: Comparison with domain randomization. Shading denotes one standard error across policies. Grey shaded areas represent ranges of training distributions for in-distribution version of domain randomization. Vertical dotted lines represent nominal training environment. Top: Total reward. Bottom: Total cost, where horizontal dotted lines represent the safety budget and values below these lines represent safety constraint satisfaction.

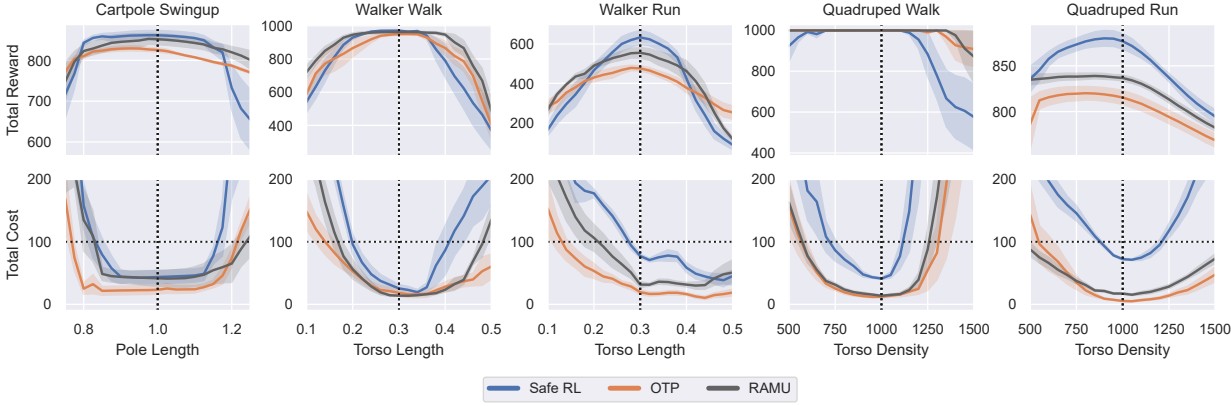

Figure 8: Comparison with distributionally robust safe RL. Shading denotes one standard error across policies. Vertical dotted lines represent nominal training environment. Top: Total reward. Bottom: Total cost, where horizontal dotted lines represent the safety budget and values below these lines represent safety constraint satisfaction.

## C.2 Comparison of safe reinforcement learning update methods

Our Optimal Transport Perturbations can be combined with many popular safe RL algorithms that perform policy updates according to (2) by replacing standard Q functions with robust Q functions as in (4). In our main experimental results, we combined our methodology with CRPO (Xu et al., 2021), which immediately switches between maximizing total reward and minimizing total cost depending on safety constraint satisfaction. To demonstrate the flexibility of our approach, in this section we also combine our OTP framework with a Lagrangian-based safe RL update method (Tessler et al., 2019b; Ray et al., 2019). Lagrangian-based methods update the policy based on a Lagrangian relaxation of (2) given by

$$\min_{\lambda \geq 0} \max_{\pi} \; \mathbb{E}_{s \sim \mathcal{D}} \left[ \mathbb{E}_{a \sim \pi(\cdot|s)} \left[ Q_{p,r}^{\pi_k}(s,a) \right] \right] - \lambda \left( \mathbb{E}_{s \sim \mathcal{D}} \left[ \mathbb{E}_{a \sim \pi(\cdot|s)} \left[ Q_{p,c}^{\pi_k}(s,a) \right] \right] - B \right), \qquad (20)$$

where the Lagrange dual parameter $\lambda$ is updated at a slower scale throughout training. Similarly, this approach can be applied to the robust and safe policy update in (4). We apply this update method with a

Table 6: Comparison of safe RL update methods

| Algorithm | Training % Safe[§] | Deploy % Safe[†] | Normalized Ave.[‡] Reward | Cost |
|---|---|---|---|---|
| CRPO | | | | |
| Safe RL | 97%* | 51%* | 1.00* | 1.00* |
| **OTP** | **98%** | **87%** | **1.06** | **0.34** |
| Lagrange | | | | |
| Safe RL | 85%* | 62%* | 1.02 | 0.83* |
| **OTP** | **91%** | **92%** | **1.05** | **0.25** |

[§] Percentage of time throughout training that policies satisfy the safety constraint in the nominal training environment, aggregated across all tasks.

[†] Percentage of policies that satisfy the safety constraint across all tasks and test environments.

[‡] Normalized relative to the average performance of standard safe RL using CRPO for each task and test environment.

* Statistically significant difference ($p < 0.05$) compared to OTP using a paired t-test.

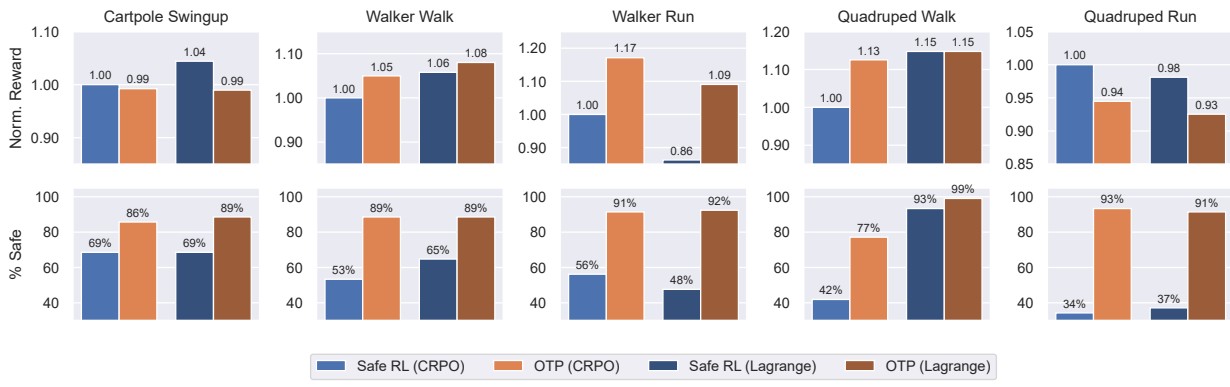

Figure 9: Performance summary by task for safe RL and OTP using different safety methods (CRPO and Lagrange), aggregated across test environments. Top: Average total reward, normalized relative to the average performance of standard safe RL using CRPO for each test environment. Bottom: Percentage of policies that satisfy the safety constraint across all test environments.

dual learning rate of $5e-6$ to both standard safe RL and our OTP framework. In Table 6 and Figure 9, we compare these experimental results to the same algorithms using CRPO. We also provide detailed results for the Lagrangian-based implementations across tasks and test environments in Figure 10.

Importantly, we see the benefits of our OTP framework across both choices of update method (CRPO and Lagrange) in terms of robust performance and safety. In both cases, our OTP framework increases the average total reward across unseen test environments at deployment time compared to standard safe RL, and leads to a significant improvement in safety at deployment time. As shown in Figure 9, the performance difference between standard safe RL and OTP is very similar across update methods. The only meaningful difference occurs in Quadruped Walk, where the Lagrangian-based approach demonstrates stronger performance and safety. However, the slight improvements shown by the Langrangian-based updates at deployment time may be the result of increased unsafe exploration during training. Because the Lagrange dual parameter $\lambda$ is updated at a slow scale throughout training, the Lagrangian relaxation in (20) can result in safety constraint violations during training. We see in Table 6 that CRPO achieves higher levels of safety during

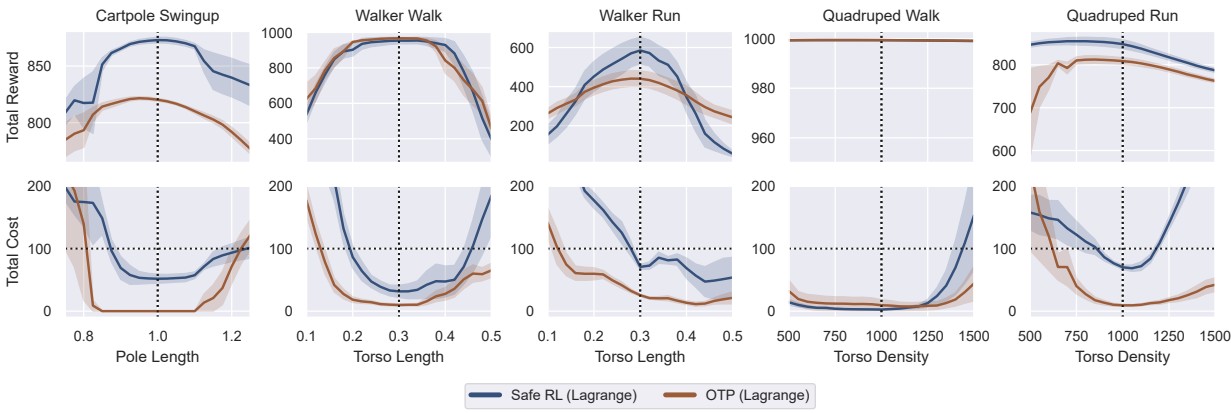

Figure 10: Comparison of OTP with standard safe RL across tasks and test environments using a Lagrangian-based update method. Shading denotes one standard error across policies. Vertical dotted lines represent nominal training environment. Top: Total reward. Bottom: Total cost, where horizontal dotted lines represent the safety budget and values below these lines represent safety constraint satisfaction.

the training process compared to a Lagrangian-based update method, as CRPO quickly reacts to safety constraint violations during training by directly minimizing total cost. Therefore, in settings where safety is also important throughout the training process, CRPO may be preferred over a Langrangian-based approach.

