# OpenReview forum: "Optimal Transport Perturbations for Safe Reinforcement Learning with Robustness Guarantees"
_TMLR — Accepted by TMLR_

### Review · Reviewer_dhZ4 · 2024-01-20

**Summary Of Contributions:**

The paper deals with safe and robust RL. However, safety is not understood here as in safe policy improvement, but in the sense of avoiding constraints.
Continuous state and action spaces are considered and experiments are conducted on five deterministic benchmarks. The method is compared with a reference method and four other variants and shows good results.

**Audience:**

Yes

**Claims And Evidence:**

Yes

**Requested Changes:**

It bothers me that there are several statements in the introduction without a reference. In my opinion, a reference should be given for each statement as far as possible, or if it is an own finding, it should be pointed out where in the paper this finding is presented. I am referring to the sentences:

> Deep reinforcement learning (RL) is a data-driven framework for sequential decision making that has demonstrated the ability to solve complex tasks, and represents a promising approach for improving real-world decision making.

> As a result, techniques have been developed to incorporate both robustness and safety into deep RL.

> Robust RL methods protect against worst-case transition models in an uncertainty set, while safe RL methods incorporate safety constraints into the training process.


Likewise, I would expect `Parametric methods assume the ability to generate a range of training environments with a detailed simulator, while adversarial methods directly influence the data collection process by attempting to negatively impact performance` to provide references to methods that are meant by these statements.


I find statement `These perturbations can be added to the training process of any safe RL algorithm to incorporate robustness to unknown disturbances, without harming performance during training or requiring access to a range of simulated training environments.` very daring. Can this really be claimed for EVERY method?
Later it is written more harmlessly: `Our Optimal Transport Perturbations can be combined with many popular safe RL algorithms`

Shouldn't a reference to safe exploration be given at `A separate line of work focuses on the issue of safe exploration during data collection`, e.g. [1]?

I find it unusual and confusing that only half a standard error is shown in Figure 4. Why?

[1] A. Hans el al., Safe Exploration in reinforcement learning, 2008

**Strengths And Weaknesses:**

**Strengths**
* The paper is carefully prepared
* The approach of using optimal transport perturbations is interesting.

**Weaknesses**
* Only deterministic benchmarks are used without mentioning this limitation.
* It remains unclear what a real application of the technique would look like.

---

> ### Author Response · Authors · 2024-02-11
> **Author Response to Reviewer dhZ4 (1/2)**
>
> Thank you for taking the time to review our paper. We appreciate your comments on the novelty and careful presentation of our methodology. Please see below for responses to your questions. We have also updated the paper to reflect your suggestions (changes are highlighted in blue).
>
> ---
>
> > **[W1] Only deterministic benchmarks are used without mentioning this limitation.**
>
> We have added commentary about our experimental analysis to the discussion on limitations and future work in Section 9 (Conclusion and Broader Impacts). Please note that our approach can be applied to both deterministic and stochastic transition models (as described after Definition 2 in Section 4), and this flexibility is a benefit of our approach. For our experiments, we consider a popular set of continuous control tasks with safety constraints that are commonly used in the literature.
>
> ---
>
> > **[W2] It remains unclear what a real application of the technique would look like.**
>
> Unlike many existing approaches to robustness in deep RL, our framework only requires data collection from a single nominal environment (real or simulated) during training. Therefore, it can be applied across a broad range of practical tasks, including settings that require real-world data collection for training.
>
> For a specific example, we believe our framework can be useful for robust, safe learning in a variety of real-world robotics applications. If a simulator is available, our OTP method can be used to learn safe policies that are robust to general environment disturbances for better sim-to-real transfer. In addition, our OTP method can be combined with real-world data collection for robust and safe fine-tuning, or to learn a robust, safe policy using only real-world data collection in complex robotics applications where accurate simulators are not currently available (e.g., contact-rich robotics with tactile feedback). This represents an interesting avenue for future work involving real-world applications, and we have added commentary about this potential application in Section 9 (Conclusion and Broader Impacts).

---

> > ### Author Response · Authors · 2024-02-11
> > **Author Response to Reviewer dhZ4 (2/2)**
> >
> > ---
> >
> > > **[RC1] It bothers me that there are several statements in the introduction without a reference.**
> >
> > Per your suggestion, we have included additional references (see below). We believe that these are high-level statements introducing the problem setting, and we have provided references throughout the paper when these concepts are discussed in more detail.
> >
> > `Deep reinforcement learning (RL) is a data-driven framework for sequential decision making that has demonstrated the ability to solve complex tasks, and represents a promising approach for improving real-world decision making.`
> >
> > * We have added a reference. There is a vast literature on deep RL and its potential applications, so we do not believe this is a controversial statement.
> >
> > `As a result, techniques have been developed to incorporate both robustness and safety into deep RL.`
> >
> > `Robust RL methods protect against worst-case transition models in an uncertainty set, while safe RL methods incorporate safety constraints into the training process.`
> >
> > * We have added standard references here for Robust MDPs and Constrained MDPs, respectively, which also appear in Sections 2 and 3. Please refer to Section 2 (Related Work), where we provide a detailed discussion of robust RL and safe RL with numerous references to popular deep RL approaches.
> >
> > `Parametric methods assume the ability to generate a range of training environments with a detailed simulator, while adversarial methods directly influence the data collection process by attempting to negatively impact performance.`
> >
> > * We have included references to parametric and adversarial methods in the paragraph immediately preceding this statement when they were first introduced. We have also provided additional details with references in Section 2 (Related Work).
> >
> > `A separate line of work focuses on the issue of safe exploration during data collection.`
> >
> > * We have added a reference. This sentence also introduces a paragraph devoted to safe exploration, where we provide several references to specific safe exploration approaches.
> >
> > ---
> >
> > > **[RC2] I find statement `These perturbations can be added to the training process of any safe RL algorithm to incorporate robustness to unknown disturbances, without harming performance during training or requiring access to a range of simulated training environments.` very daring. Can this really be claimed for EVERY method? Later it is written more harmlessly**
> >
> > We have updated the wording to ` These perturbations can be added to the training process of many popular safe RL algorithms…`, which matches the wording used later in the paper.
> >
> > ---
> >
> > > **[RC3] I find it unusual and confusing that only half a standard error is shown in Figure 4. Why?**
> >
> > We do not believe this is unusual in the deep RL literature, but we would be happy to update the shading in the figures. We have updated the figures to show one standard error which we hope addresses your concern.

---

### Review · Reviewer_qng5 · 2024-01-30

**Summary Of Contributions:**

This paper proposes a novel way of quantifying uncertainty sets in deep RL, for improved robustness and safety. The key contribution of the paper is to define uncertainty sets at each time-step of a control problem, based on an optimal transport cost from a nominal state-transition model. The benefit of this uncertainty set quantification that data only needs to be collected in a nominal environment. Results across simulated mujoco environments show that the proposed approach can be incorporated with an existing safe RL algorithm to obtain lower costs compared to the baseline, and improved robustness compared to standard domain randomization.

**Audience:**

Yes

**Claims And Evidence:**

Yes

**Requested Changes:**

Changes necessary for acceptance

- please provide a detailed discussion (ideally with theorems) of quantifying safety / robustness of the approach. Currently, there isn't enough discussion about safety guarantees for the proposed approach - how much is worst case safety or average case safety improved compared to a baseline?

- please provide experimental justifications for "Our Optimal Transport Perturbations can be combined with many popular safe RL algorithms, which is a benefit of our methodology" or quantify for what type of safe RL algorithms (for example those discussed in the related works) it will be non-trivial to apply the proposed approach


Changes not necessary for acceptance

- it will be good to experiment with slightly more complex control problems (ref to weakness above)

**Strengths And Weaknesses:**

Strengths:

- the proposed approach for uncertainty set quantification is simple to implement, and the paper shows how it can be added on top of one safe RL algorithm

- the details in the paper and appendix are clear, and provide sufficient understanding of the implementation details in the experiments.

- a major strength of the approach is that it doesn't require a simulator (unlike domain randomization) for improving robustness, and so in principle can be incorporated for hard to simulate real-world control problems

- the paper is overall easy to follow, and the theorems introduced in the paper have clear assumptions, and detailed proof in the appendix.

Weaknesses

- the paper's experiment section claims "Our Optimal Transport Perturbations can be combined with many popular safe RL algorithms, which is a benefit of our methodology" however the results are shown only with respect to ONE existing safe Rl algorithm. So this claim is not fully substantiated.

- the DM control simulation environments are very simple control problems and it is unclear whether the approach can be applied for improving safety (or robustness) in more complex and high dimensional problems (for example robot manipulation). In particular, it is unclear whether the approach significantly improves upon the baseline in terms of behavioral changes (e.g. the baseline is not able to solve a task at all i.e. ~0% success rate due to safety failures, while addition of the approach improves success by > 20%)

- there isn't enough discussion about safety guarantees for the proposed approach - how much is worst case safety or average case safety improved compared to a baseline?

---

> ### Author Response · Authors · 2024-02-11
> **Author Response to Reviewer qng5 (1/2)**
>
> Thank you for taking the time to review our paper. We are glad you found the paper clear and detailed, and we are excited that you see the broad potential of our framework due to its use of data from a single nominal training environment. Please see below for responses to your questions. We have also updated the paper to reflect your suggestions (changes are highlighted in blue).
>
> ---
>
> > **[W1] the paper's experiment section claims `Our Optimal Transport Perturbations can be combined with many popular safe RL algorithms, which is a benefit of our methodology` however the results are shown only with respect to ONE existing safe RL algorithm. So this claim is not fully substantiated.**
>
> > **[RC2] please provide experimental justifications for `Our Optimal Transport Perturbations can be combined with many popular safe RL algorithms, which is a benefit of our methodology` or quantify for what type of safe RL algorithms (for example those discussed in the related works) it will be non-trivial to apply the proposed approach**
>
> When comparing the robust and safe RL update in (4) to the standard safe RL update in (2), the only difference is the use of robust Q functions instead of standard Q functions. Therefore, we can combine our methodology with any safe RL algorithm that approximately implements the update in (2) by replacing standard Q functions with our robust Q functions (as in the update in (4)). We have updated the wording of this claim in the Experiments section to make this more clear. Note that our work primarily focuses on how to efficiently learn the robust Q functions that appear in (4) for an optimal transport uncertainty set.
>
> We view the choice of safe RL algorithm as an implementation detail that determines how to approximately implement the policy updates in (2) and (4), and for a fair comparison we applied the same baseline safe RL algorithm in every method we considered in our experiments. In our experiments, we demonstrated the benefits of our methodology for the popular safe RL algorithm CRPO [1]. In response to your suggestion, we have also combined our methodology with a Lagrangian-based safe RL update method to demonstrate another implementation of our OTP framework. This is another popular approach to approximating the safe RL update in (2), as discussed in the Related Work section. Please see Section C.3 for these experimental results, which demonstrate similar trends to the main experimental results using CRPO.
>
> [1] Xu et al. CRPO: A new approach for safe reinforcement learning with convergence guarantee. In Proceedings of the 38th International Conference on Machine Learning, pp. 11480–11491. PMLR, 2021.
>
> ---
>
> > **[W2] the DM control simulation environments are very simple control problems and it is unclear whether the approach can be applied for improving safety (or robustness) in more complex and high dimensional problems (for example robot manipulation).**
>
> > **[RC3] Changes not necessary for acceptance: it will be good to experiment with slightly more complex control problems (ref to weakness above)**
>
> The Real-World RL Suite [2, 3] is a popular set of continuous control tasks that incorporate safety constraints and environment perturbations, and this set of benchmark tasks was designed to test robustness and safety in RL (the focus of our work). These tasks vary in dimensionality (cartpole: $|\mathcal{S}|= 5, |\mathcal{A}| = 1$, walker: $|\mathcal{S}|= 24, |\mathcal{A}| = 6$, quadruped: $|\mathcal{S}|= 78, |\mathcal{A}| = 12$), and are not trivial to solve given the presence of safety constraints. We believe these tasks are sufficiently complex to evaluate the robustness benefits of our framework, as evidenced by the fact that standard safe RL only remains safe in 51% of the test environments we considered. In addition, because it is possible to achieve strong, safe performance in the nominal training environment for these tasks, we can better evaluate test-time robustness in terms of both performance and safety across environments not seen during training.
>
> We agree that applications of our approach to additional tasks and problems, including real-world robotics applications, is an interesting avenue for future work. We have added commentary about our experimental analysis to the discussion on limitations and future work in Section 9 (Conclusion and Broader Impacts).
>
> [2] Dulac-Arnold et al. An empirical investigation of the challenges of real-world reinforcement learning. arXiv preprint arXiv:2003.11881, 2020.
>
> [3] Dulac-Arnold et al. Challenges of real-world reinforcement learning: definitions, benchmarks and analysis. Machine Learning, 110:2419–2468, 2021.

---

> > ### Author Response · Authors · 2024-02-11
> > **Author Response to Reviewer qng5 (2/2)**
> >
> > ---
> >
> > > **[W3] there isn't enough discussion about safety guarantees for the proposed approach - how much is worst case safety or average case safety improved compared to a baseline?**
> >
> > > **[RC1] please provide a detailed discussion (ideally with theorems) of quantifying safety / robustness of the approach. Currently, there isn't enough discussion about safety guarantees for the proposed approach - how much is worst case safety or average case safety improved compared to a baseline?**
> >
> > We have added Remark 1 after (3) to emphasize the guarantees associated with the Robust Constrained MDP (RC-MDP) framework. Our work considers the RC-MDP framework, so our method achieves these guarantees where $\mathcal{P}$ is an optimal transport uncertainty set. We have also added sentences in Section 3.2 and Section 5 to emphasize that we are focused on solving the robust and safe RL problem given by (3).
> >
> > We have also provided detailed experimental analysis that demonstrates the safety benefits of our algorithm. See the "% Safe" column in Table 1 and the bottom rows of Figures 3 and 4 in the Experiments section. Due to the guarantees associated with the RC-MDP framework discussed above, our OTP method remains safe in 87% of test cases while standard safe RL remains safe in only 51% of test cases. In addition, our algorithm achieves this robust safety while using the same data collection process as standard safe RL.

---

> > > ### Comment · Reviewer_qng5 · 2024-02-27
> > > **thanks for the response**
> > >
> > > Dear authors - thanks for clarifying the concerns of all the reviewers! My main requested clarifications / edits are resolved.

---

### Review · Reviewer_vzU7 · 2024-02-03

**Summary Of Contributions:**

This paper tackles the robustness problem in safe reinforcement learning. The main contribution is that the authors propose to use the optimal transport uncertainty set, leading to an efficient implementation of the worst-case optimization problems. This paper provides both a theoretical analysis of the uncertainty set and an empirical implementation of a deep RL algorithm. Specifically, the empirical implementation introduces an additional component named Optimal Transport Perturbations to generate the perturbations.

**Audience:**

Yes

**Broader Impact Concerns:**

There is no ethical concern as far as I know.

**Claims And Evidence:**

No

**Requested Changes:**

1. Add more experiments on other benchmarks that are designed for Safe RL.
2. Add more recent baselines as discussed in the weakness section.
3. The authors need to explain the reason for sharing a similar structure and content with [1].

[1] Risk-Averse Model Uncertainty for Distributionally Robust Safe Reinforcement Learning

**Strengths And Weaknesses:**

**Strengths:**

1.	Using optimal transport as uncertainty set in safe RL seems novel.
2.	The writing flow of the paper is good, which provides the necessary content to help me understand the high-level idea and details.
3.	The empirical implementation seems to be highly related to the theoretical analysis.

**Weaknesses:**
1. It seems that the proposed method uses different hyperparameters for different tasks. This might be a signal of weak generalizability or stability of the proposed method. An empirical analysis of the influence of the hyper-parameters (e.g., safety coefficient) could be more convincing.
2. Missing related works and baselines: Liu, Zuxin, et al. "Towards robust and safe reinforcement learning with benign off-policy data." International Conference on Machine Learning. PMLR, 2023.
3. Since this paper focuses on safe RL, I assume that the experiments should be run on tasks that are highly related to risks, such as safety-gym or self-driving. The safety constraints used in this paper are mainly about joint angle and joint velocity, which may not be consistent with the binary cost setting in the formulation and are not very critical in real-world settings.
4. This paper seems to share very similar content and structure with [1] in related work, problem formulation, empirical implementation, and experiments. Since these two papers propose fundamentally different methods, I am worried that there are too many overlaps between them. In addition, I don’t find [1] as a baseline in the experiments.

[1] Risk-Averse Model Uncertainty for Distributionally Robust Safe Reinforcement Learning

---

> ### Author Response · Authors · 2024-02-11
> **Author Response to Reviewer vzU7 (1/2)**
>
> Thank you for taking the time to review our paper. We appreciate your comments on the novelty, clear presentation, and empirical implementation of our algorithm. Please see below for responses to your questions. We have also updated the paper to reflect your suggestions (changes are highlighted in blue).
>
> We strongly believe that all of the claims made in the paper are technically sound and supported by clear evidence. We hope that our responses and paper revisions have addressed any concerns you may have about this. If so, we ask that you please consider updating your “Claims and Evidence” response to Yes to reflect these clarifications and revisions. We would also be happy to provide additional clarifications on any claims that you believe are not supported by evidence.
>
> ---
>
> > **[W1] It seems that the proposed method uses different hyperparameters for different tasks. This might be a signal of weak generalizability or stability of the proposed method. An empirical analysis of the influence of the hyper-parameters (e.g., safety coefficient) could be more convincing.**
>
> Our proposed OTP algorithm uses the same hyperparameter settings across every task (see Table 5 for details). This is another benefit of our approach compared to parametric methods such as domain randomization, which must consider a task-specific definition of uncertainty.
>
> Your question may be referring to the definitions of safety constraints in each task, which are characteristics of the tasks and **not** an algorithm hyperparameter. We consider the same safety constraints used in prior work [1]. These safety constraints all consider the same safety budget $B$ and were selected in [1] to be the most difficult safety constraint to satisfy (i.e., smallest safety coefficient value in the Real-World RL Suite [2, 3]) where it is still possible to achieve strong performance in the nominal training environment. In order to better understand the impact of the safety coefficient value, please see the total cost for our learned policies across different safety coefficient values here: https://figshare.com/s/7fd47b01de5a50be1502. Note that this represents a change in the cost function under the same environment dynamics. Our work, on the other hand, focuses on robust performance and safety for a given cost function across different environment dynamics.
>
> ---
>
> > **[W2] Missing related works and baselines: Liu, Zuxin, et al. "Towards robust and safe reinforcement learning with benign off-policy data." International Conference on Machine Learning. PMLR, 2023.**
>
> > **[RC2] Add more recent baselines as discussed in the weakness section.**
>
> We have added a reference to the paper you suggested in our related work, along with another relevant paper by the same author. Please note that these works focus on robustness to *observational perturbations* in safe RL (which do not impact environment dynamics). This is different from the focus of our work, which considers robustness to *transition dynamics* in safe RL. For this reason, we believe this reference should be included in our related work but is not an appropriate baseline for our experimental analysis. Our experimental analysis compares against popular baselines that represent common implementations of robustness to transition dynamics in deep RL.

---

> > ### Author Response · Authors · 2024-02-11
> > **Author Response to Reviewer vzU7 (2/2)**
> >
> > ---
> >
> > > **[W3] Since this paper focuses on safe RL, I assume that the experiments should be run on tasks that are highly related to risks, such as safety-gym or self-driving. The safety constraints used in this paper are mainly about joint angle and joint velocity, which may not be consistent with the binary cost setting in the formulation and are not very critical in real-world settings.**
> >
> > > **[RC1] Add more experiments on other benchmarks that are designed for Safe RL.**
> >
> > The Real-World RL Suite [2, 3] is a popular set of continuous control tasks that incorporate safety constraints and environment perturbations, and this set of benchmark tasks was designed to test robustness and safety in RL (the focus of our work). We apply safety constraints that are already defined within the Real-World RL Suite.
> >
> > Note that our paper (including our experimental analysis) focuses on safety defined by a Constrained MDP (C-MDP) (see Section 3.1), which considers a safety budget $B$ on expected total discount costs (and is not restricted to a binary cost setting). This is a common definition of safety in the RL literature. We believe that our experiments provide a thorough analysis of this setting and demonstrate the benefits of our proposed approach within this setting.
> >
> > We agree that applications of our approach to additional tasks and other definitions of safety is an interesting avenue for future work. Your comment appears to focus on safety-critical scenarios defined according to unsafe states (different from the C-MDP setting we consider), which we have mentioned in our discussion of limitations and future work in Section 9 (Conclusion and Broader Impacts).  In the revised version of the paper, we have also included additional commentary about our experimental analysis in Section 9.
> >
> > ---
> >
> > > **[W4] This paper seems to share very similar content and structure with [1] in related work, problem formulation, empirical implementation, and experiments. Since these two papers propose fundamentally different methods, I am worried that there are too many overlaps between them. In addition, I don’t find [1] as a baseline in the experiments.**
> >
> > > **[RC3] The authors need to explain the reason for sharing a similar structure and content with [1].**
> >
> > We follow the same experimental design considered in [1], which we state explicitly at the beginning of the Experiments section (including a reference to [1]). Using this experimental design, our Experiments section focuses on demonstrating the benefits of the novel OTP methodology introduced in this work, which also incorporates additional analysis and baseline comparisons compared to [1].
> >
> > Our paper investigates similar high-level concepts as [1] (i.e., robustness and safety in RL), so it is logical that each paper would share some references in related works. However, as you have mentioned, **the core novel contributions of these works are very different**. Our work and [1] consider different notions of robustness that lead to **different problem formulations**: our work focuses on *robust RL* defined using uncertainty sets of transition models, while [1] is equivalent to *distributionally robust RL* that applies robustness around a *distribution over transition models* (not a single nominal transition model). Therefore, our work provides stronger robustness guarantees related to worst-case transition models in $\mathcal{P}$ compared to the (weaker) distributionally robust guarantees of [1]. As expected, this leads to methods and theoretical results that are also very different.
> >
> > Because these works consider different problem formulations (robust vs. distributionally robust), we did not include the algorithm from [1] as a method for comparison in our experimental analysis. Based on your suggestion, we have included a comparison with [1] in Section C.2. We discuss the difference in problem formulations described above, and we provide an experimental comparison that provides additional insight into these differences.
> >
> > ---
> >
> > **References:**
> >
> > [1] Queeney and Benosman. Risk-averse model uncertainty for distributionally robust safe reinforcement learning. In Advances in Neural Information Processing Systems, volume 36. Curran Associates, Inc., 2023.
> >
> > [2] Dulac-Arnold et al. An empirical investigation of the challenges of real-world reinforcement learning. arXiv preprint arXiv:2003.11881, 2020.
> >
> > [3] Dulac-Arnold et al. Challenges of real-world reinforcement learning: definitions, benchmarks and analysis. Machine Learning, 110:2419–2468, 2021.

---

> > > ### Comment · Reviewer_vzU7 · 2024-02-21
> > > **Response**
> > >
> > > Thanks for the authors providing more explanation and experiments. All of my concerns have been resolved. One thing I kindly disagree with is that even if the formulation between this work and [1] are different, they are still comparable if they can be tested on the same benchmark.
> > >
> > > From what I understand, the difference mainly comes from how the algorithm estimates the noise, which is not the fundamental difference of the testing environment (I assume this paper uses the same environment as in [1]). If so, I highly suggest the authors put the results of [1] in Table 1 and add a footnote to indicate that these two belong to different types of algorithms. It would benefit other researchers when they want to use this benchmark and decide which type of algorithm they want to improve.
> > >
> > > Such a comparison also raises other interesting questions: if these two methods achieve similar results, what's the additional benefit of using a distributionally robust framework rather than a robust framework? Can this benchmark support the advantage?

---

> > > > ### Author Response · Authors · 2024-02-26
> > > > **Author Follow-Up Response to Reviewer vzU7**
> > > >
> > > > We are glad that we have been able to resolve all of your concerns. In response to your suggestion, we have moved the comparison with [1] from Section C into the main experiments section (Section 8). We have included the results from [1] in Table 1 and Figure 3, and we have included a discussion of the key differences between these approaches in Section 8.4. Notably, our framework achieves statistically significant improvements in safety constraint satisfaction compared to [1] due to its stronger robustness guarantees.

---

### Author Response · Authors · 2024-02-11
**Author Response to Reviewers**

Thank you to all of the reviewers for their thoughtful feedback. We are glad the reviewers agree that the paper contributes a novel and interesting method for robustness and safety in deep RL. We have replied directly to each reviewer with detailed responses, and we have updated the paper to incorporate additional clarifications, discussion, and experimental analysis based on reviewer suggestions (changes are highlighted in blue). Thank you for helping us to improve our paper!

---

### Decision · Action_Editor_UVgP · 2024-03-15

**Recommendation:** Accept as is

**Comment:**

qng5: "The paper has some interesting ideas like proposing a novel way of quantifying uncertainty sets in deep RL. After revision, most of the claims are well-quantified in the paper and I do not see any major issues with the work. The experimental results are not as exciting (simple DM control environments) and there is no experimental validation of combining the proposed optimal transport perturbations approach with multiple safe RL algortihms (this claim is only justified through arguments, not through experiments). However, despite these limitations, I think the paper meets the TMLR bar for acceptance."

vzU7: "All my concerns have been resolved in the response of the authors."

dhZ4: "My objections were minor or stylistic in nature and have now all been addressed by the authors, so that it is now a solid paper on an interesting topic."

**Audience:**

Safety and robust control, reinforcement learning, and robotics communities would be infestered in this work.

**Claims And Evidence:**

The paper addresses the robustness and safety in RL through uncertainty sets quantification, improving the worst case optimization problems. One of the strengths of the method is that it can be seamlessly integrated into the an existing safe RL algorithms to make them more computationally efficient.

The reviewers are in the agreement that the method is elegant, well written, and offers both empirical and theoretical analysis. The reviewers appreciate the revised version.